# CVQA: Culturally-diverse Multilingual Visual Question Answering Benchmark

**David Romero**[*♣], **Chenyang Lyu**[*♣], **Haryo Akbarianto Wibowo**[♣], **Teresa Lynn** , **Injy Hamed** ,
**Aditya Nanda Kishore** , **Aishik Mandal** , **Alina Dragonetti** , **Artem Abzaliev** ,
**Atnafu Lambebo Tonja** , **Bontu Fufa Balcha** , **Chenxi Whitehouse** , **Christian Salamea** ,
**Dan John Velasco** , **David Ifeoluwa Adelani** , **David Le Meur** , **Emilio Villa-Cueva** ,
**Fajri Koto** , **Fauzan Farooqui** , **Frederico Belcavello** , **Ganzorig Batnasan** , **Gisela Vallejo** ,
**Grainne Caulfield** , **Guido Ivetta** , **Haiyue Song** , **Henok Biadglign Ademtew** , **Hernán Maina** ,
**Holy Lovenia** , **Israel Abebe Azime** , **Jan Christian Blaise Cruz** , **Jay Gala** , **Jiahui Geng** ,
**Jesus-German Ortiz-Barajas** , **Jinheon Baek** , **Jocelyn Dunstan** , **Laura Alonso Alemany** ,
**Kumaranage Ravindu Yasas Nagasinghe** , **Luciana Benotti** , **Luis Fernando D'Haro** ,
**Marcelo Viridiano** , **Marcos Estecha-Garitagoitia** , **Maria Camila Buitrago Cabrera** ,
**Mario Rodríguez-Cantelar** , **Mélanie Jouitteau** , **Mihail Mihaylov** , **Naome Etori** ,
**Mohamed Fazli Mohamed Imam** , **Muhammad Farid Adilazuarda** , **Munkhjargal Gochoo** ,
**Munkh-Erdene Otgonbold** , **Olivier Niyomugisha** , **Paula Mónica Silva** , **Pranjal Chitale** ,
**Raj Dabre** , **Rendi Chevi** , **Ruochen Zhang** , **Ryandito Diandaru** , **Samuel Cahyawijaya** ,
**Santiago Góngora** , **Soyeong Jeong** , **Sukannya Purkayastha** , **Tatsuki Kuribayashi** ,
**Teresa Clifford** , **Thanmay Jayakumar** , **Tiago Timponi Torrent** , **Toqeer Ehsan** ,
**Vladimir Araujo** , **Yova Kementchedjhieva** , **Zara Burzo** , **Zheng Wei Lim** , **Zheng Xin Yong** ,
**Oana Ignat** , **Joan Nwatu** , **Rada Mihalcea** , **Thamar Solorio**[♣], and **Alham Fikri Aji**[♣]

♣Core Authors (MBZUAI)
www.cvqa-benchmark.org

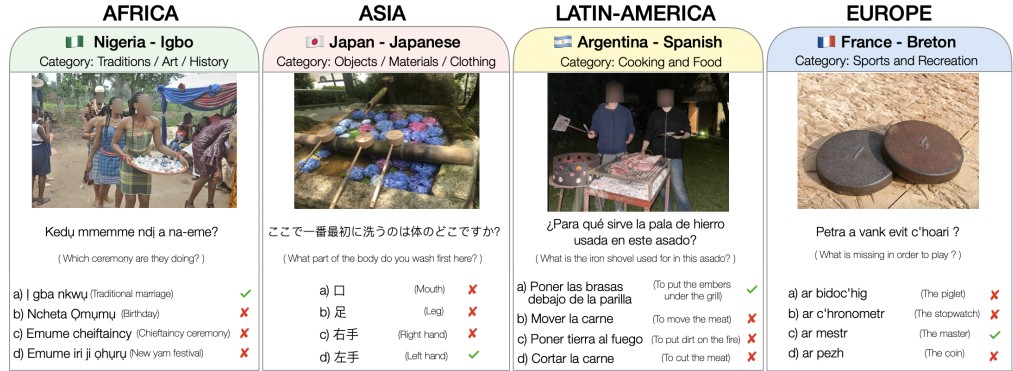

Figure 1: We propose CVQA, a large-scale multilingual VQA benchmark, representing the cultures of 30 countries and 31 different languages across 10 diverse categories, comprising 10k samples.

## Abstract

Visual Question Answering (VQA) is an important task in multimodal AI, and it is often used to test the ability of vision-language models to understand and reason on knowledge present in both visual and textual data. However, most of the current VQA models use datasets that are primarily focused on English and a few major world languages, with images that are typically Western-centric. While recent efforts have tried to increase the number of languages covered on VQA

---

[*]Equal Contribution

38th Conference on Neural Information Processing Systems (NeurIPS 2024) Track on Datasets and Benchmarks.

datasets, they still lack diversity in low-resource languages. More importantly, although these datasets often extend their linguistic range via translation or some other approaches, they usually keep images the same, resulting in narrow cultural representation. To address these limitations, we construct CVQA[2], a new **C**ulturally-diverse multilingual **V**isual **Q**uestion **A**nswering benchmark, designed to cover a rich set of languages and cultures, where we engage native speakers and cultural experts in the data collection process. As a result, CVQA includes culturally-driven images and questions from across 30 countries on four continents, covering 31 languages with 13 scripts, providing a total of 10k questions. We then benchmark several Multimodal Large Language Models (MLLMs) on CVQA, and show that the dataset is challenging for the current state-of-the-art models. This benchmark can serve as a probing evaluation suite for assessing the cultural capability and bias of multimodal models and hopefully encourage more research efforts toward increasing cultural awareness and linguistic diversity in this field.

# 1   Introduction

Visual Question Answering (VQA) [2, 43, 50] is a task that requires AI systems to answer textual questions based on a given context image. VQA serves as an essential measure for assessing the understanding and reasoning capabilities of Multimodal Large Language Models (MLLMs) across diverse images and texts. With the rapid development of MLLMs, significant improvements have been observed, including support for multiple languages [12, 5, 27, 45, 53]. However, there is still a lack of VQA benchmarks that capture a diverse set of languages and cultural contexts. Specifically, most VQA benchmarks only cover the English language [2, 33]. While some work has been undertaken on multilingual VQA, it either covers a limited set of popular languages or is producing questions via translation/generation of text from the original Western-centric images, thus failing to capture cultural nuances inherent in different languages [6, 44].

To address these limitations, we propose CVQA: a novel, large-scale, multilingual, culturally nuanced VQA benchmark that includes a diverse set of languages, including many that are underrepresented and understudied. CVQA follows the grassroots crowd-sourcing collaboration approaches taken by Masakhane for Africa [37], NusaCrowd for Indonesia [4], and AI4Bharat for India [20]. In our case, however, we collaborate across communities, rather than within one particular community, in order to maximize cultural and linguistic representation. Consequently, our data consists of 10k questions across 30 countries, covering 31 languages. We also sub-categorize CVQA based on Country-Language pairs, resulting in 39 distinct pairs, which is substantially more extensive than existing VQA benchmarks. Furthermore, each sample in CVQA falls into one of 10 diverse categories (see Table 1) and is annotated and validated by fluent speakers and those familiar with the respective cultures, ensuring high quality and diversity. Lastly, CVQA is written in both English and local languages, enabling us to benchmark multilingual MLLMs and English-only MLLMs.

In this study, we benchmark CVQA across various MLLMs and find that it presents a significant challenge for open MLLMs, which most of the time achieve no more than 50% accuracy. Additionally, we observe a notable degradation in model performance when questions are asked in native languages, particularly those in understudied languages such as Breton from France and Javanese from Indonesia, highlighting a significant gap in understanding multilingual prompts. We further conduct several ablation studies to analyze the models' performance across different question categories, regions, languages, and image sources. Our contributions can be summarised as follows:

- First, we introduce CVQA, a new culturally diverse multilingual visual question answering dataset consisting of over 10,000 questions from across 30 countries and 31 languages.

- Second, we provide extensive documentation on our process to crowdsource such large dataset across numerous communities, including annotation guidelines.

- Finally, we provide an initial set of evaluations on this benchmark, to serve as a baseline for future research on vision-language models that are culturally diverse.

---

[2]https://huggingface.co/datasets/afaji/cvqa

We note that efforts to enhance cultural awareness in models are increasingly gaining attention. As such, our work contributes to the growing interest within the community and can encourage further initiatives to broaden the limited world view currently captured by MLLMs.

## 2 CVQA Data Collection

The construction of our CVQA dataset involved a detailed annotation process that aims at creating a culturally diverse and linguistically comprehensive dataset for Visual Question Answering. It is worth noting that, while defining culture is challenging, we follow Adilazuarda et al. [1] by using common-ground knowledge (e.g., information surrounding local dishes, history, places, etc. that is generally shared by the people within the region) as a proxy of culture. In this section, we now turn to outline the detailed procedures followed during the data collection and annotation phases.

### 2.1 Dataset Collection Design

**Country-Language Pair Subset** CVQA is a multilingual, multiple-choice locally-nuanced visual question-answering dataset. The format is similar to commonly used visual QA data such as VQA [2], VQA-2 [13] or GQA [17]. Yet, in contrast to them, we gathered images and created question-answer pairs based on the cultures of various locations. Moreover, for each location, the question-answer pairs were created in their respective local languages, along with parallel English translations. Some languages are shared across different locations (e.g., Mexico-Spanish vs Spain-Spanish), and vice-versa, different languages are shared across the same location (e.g., Indonesia-Indonesian vs Indonesia-Javanese). Therefore, to capture them, we group our CVQA dataset into several subsets based on this Country-Language pair, rather than simply on language or location only.

**Annotators** To elicit image collectors and annotation contributions to this project, we reached out to our network, which included both linguistic groups and NLP communities. Annotators needed to be fluent speakers of the language in question and be accustomed to the cultures of the locations for which they provided data. To promote data collection, contributors with significant contributions, either by contributing at least 100 validated question-answer pairs and/or managing several subsets, are rewarded as co-authors in this paper. The annotator demographic statistics can be seen in Figure 8, Appendix D. Our annotators are predominantly native speakers, with around 89% residing in the respective country for over 16 years. The age group distribution shows a significant concentration in the 18-30 age bracket, with about one-third female representation. Overall, the demographic profile highlights diversity in terms of age, with high levels of cultural familiarity and language proficiency.

**Categories** For the categorization of questions of our CVQA dataset, we incorporate 10 diverse categories to ensure a culturally-comprehensive representative set of visual questions, which are shown in Table 1. We mainly adopt the categorization from the OK-VQA dataset [33], with some modifications to fit the theme of our project. Specifically, the categories from the OK-VQA dataset used in our CVQA dataset are 1) to 7). We split the original category of *Geography, History, Language and Culture* into 2 separate categories of 8) and 9). In addition, we added a new category of 10) considering the effect that cultural icons and media have on everyday life.

Table 1: Categories in our Dataset. To save space in some of our results, we might refer them by shorthand version in brackets.

| Category |
| --- |
| 1. Vehicles and Transportation (Vehicles) |
| 2. Cooking and Food (Food) |
| 3. People and Everyday Life (People) |
| 4. Sports and Recreation (Sports) |
| 5. Plants and Animals (Plants & Animals) |
| 6. Objects, Materials, and Clothing (Objects) |
| 7. Brands and Products (Brands) |
| 8. Geography, Buildings, and Landmarks (Geography) |
| 9. Tradition, Art, and History (Tradition) |
| 10. Public Figure and Pop-Culture (Pop Culture) |

### 2.2 Annotation Process

We developed concise annotation guidelines (in English) that are suitable for all Country-Language subset teams. Here we provide an overview of the key steps that annotators followed during the dataset creation process. The full guidelines are provided in Appendix A.

**Image Selection and Preparation** For each Country-Language pair, annotators were instructed to select images that depict diverse cultural aspects pertinent to their cultural backgrounds among one of

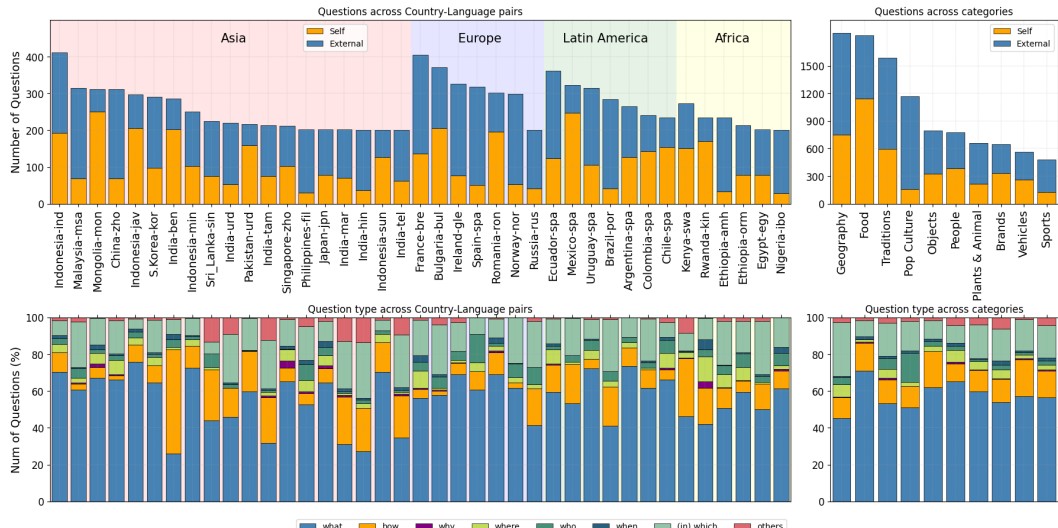

Figure 2: Statistics of the CVQA Benchmark

the 10 categories. We did not enforce balance across categories considering the different variations of cultural knowledge. We strongly recommend that annotators use their own personal images to avoid accidental data leakage from existing online sources. However, we noted that this request was not always possible, since some images are extremely hard to come by (e.g., photos of public figures or landmarks that are far from the annotator's location). Therefore, we also allowed them to use images from our pre-defined list of open-use licensing sources[3]. For self-made images, we asked the annotators whether they were willing to make the image available for commercial or research purposes. For images from existing online sources, we applied the original license.

We requested annotators to avoid using sensitive images that would perpetuate stereotypes. In addition, the annotators were also requested to anonymize faces that were not public figures or fictional characters, as well as text that could reveal the answer to the accompanying questions. We also post-processed all images to remove all metadata such as geo-location, device type, and so on.

**Question Creation**    The questions associated with each image had to be culturally relevant and formulated such that they would require the context of the image in order to be answerable. A maximum of three question-answer pairs could be provided for each image. Each question was accompanied by one correct answer and three distractors that were reasonably plausible, yet incorrect, thus forming a multiple-choice format.

While we follow the existing VQA benchmarks in terms of using a multiple-choice format, we are also aware that multiple-choice has some flaws when used to measure a model's performance [41]. Hence, we made sure that CVQA is also convertible into free-text open-ended QA, by instructing the annotators to ensure that the question would be answerable even without the accompanying multiple choices (i.e., not through a deductive method). Moreover, to accommodate the multilingual aspect of the benchmark, each question-answer pair was created in the local language and manually translated into English.

Annotators were advised to create questions that promoted an understanding and appreciation of different cultures without perpetuating stereotypes. Typical questions ranged from simple identification queries (e.g., "What is the name of this food?") to more complex ones involving multi-hop reasoning or local common-sense knowledge (e.g., "What is the color of the t-shirt the youngest member of this group is wearing?").

**Annotation Examples and Training**    The annotation guidelines provided multiple examples of well-formulated questions and answers to help guide annotator efforts (See Appendix A). These examples helped clarify the level of specificity and cultural relevance expected in the annotations.

---

[3]common.wikimedia.org, Flickr, GapMinder, Unsplash, Pixabay

We provided a tutorial to annotators on how to edit and blur sensitive information in the images. To confirm understanding, we spot-checked the annotators' collected data throughout the annotation period and informed them if some of their data did not follow the guidelines.

**Validation** The last step in the CVQA data creation was the validation process. Each entry was validated by another annotator of the same Country-Language pair. The validators were instructed to ensure that each question followed the guidelines. Based on our spot-checking, common mistakes that we encouraged the validators to check were typos and grammatical mistakes, non-cultural questions, questions that could be answered without the image, as well as incorrectly-sourced images. More information on the annotation platform is provided in Appendix B.

## 2.3 Data Statistics

To ensure sufficient question variation, we set the minimum number of questions to be included in CVQA to be at least 200 questions per Country-Language subset. In the end, we gathered 10,374 total questions across all subsets. Some statistics of our collected data are shown in Table 2. Our CVQA covers a diverse set of languages and locations spread across the globe. We also capture languages written in various scripts. While Latin is the dominant script (used in 22 Country-Language pairs), the remaining scripts are diverse; covering Arabic, Amharic, Bengali, Chinese, Cyrillic, Devanagari, Hangul, Japanese, Perso-Arabic, Sinhalese, Tamil, and Telugu. The Country-Language pairs and corresponding scripts are shown in Appendix E. CVQA covers several less commonly studied languages and regions, such as Ireland-Irish, Indonesia-Minangkabau, Indonesia-Javanese, France-Breton, Nigeria-Igbo and Mongolia-Mongolian.

Question distribution across the subset and categories are shown in Figure 2. Whether the image is coming from an external or personal source varies depending on the subset. We also note that the category with the most personal images is *Cooking and Food*, which we assume is due to the ease of obtaining such images. In contrast, the category with the least amount of personal images is *Public Figures and Pop Culture*, as it is less likely for people to have personal photos under this category.

To investigate the question variations, we categorize each question into question types of "what", "how", "why", "where", "who", and "which" questions. We categorize the questions by simple string-matching performed on the English questions. While not perfect, we argue that this method should be able to capture the trend of the questions. As shown in Figure 2, the majority of the questions fall into "what" questions. Question distribution across different Country-Language pairs varies, with an interesting finding that India-Bengali has a lot of "how" questions. Across categories, perhaps unsurprisingly, the *Geography*

Table 2: CVQA Data Statistics

| | |
|---|---|
| No. of images | 5,239 |
| No. of questions | 10,374 |
| No. of countries | 30 |
| No. of languages | 31 |
| No. of country-language pairs | 39 |
| Avg. questions per image | 1.98 |
| Avg. words per question | 7.6 |
| Avg. words per option | 1.80 |

*and Landmark* category has noticeably more "where" and "which" (e.g., in which city) questions, whereas the *Public Figure and Pop Culture* category has more "who" questions. By looking at the most frequently used words (Figure 7) across each category, we can see the general theme of the types of questions being asked. For example, questions in the *Cooking and Food* category often enquire about dish names, ingredients, or tastes.

## 3 Experimental Setup

**Models.** To evaluate performance on our CVQA benchmark, we select a range of multimodal vision-language models with multilingual and monolingual English-only capabilities. For monolingual English-only models, we test CLIP [40] a contrastive-learning-based model, trained with approximately 400 million images and English-only text pairs from the web, where we use its *vit-large-patch14-336* version. We also use InstructBLIP(4.1B) [8], an English-only instruction-aware vision model based on BLIP-2 [24], trained with 13 held-in datasets covering different tasks in English. For multilingual models, we evaluate LLaVA-1.5 (7B) [22] based on Llama-2 [46], and mBLIP [12] a BLIP-2 based model that covers 96 languages (where we evaluate two model variations, mBLIP mT0-XL (4.9B) and mBLIP BLOOMZ (8.3B)). Lastly, we employ M-CLIP [5] a multilin-

gual CLIP-based model that supports 68 languages, where we use its *XLM-Roberta-Large-Vit-B-32* version. We also evaluate the most advanced closed-source MLLMs, such as GPT-4o [36] and Gemini-1.5-Flash [45].

**Evaluation Framework.** We perform a zero-shot evaluation with two types of prompts, as follows: a location-aware prompt, which specifies the country, the question, and the options, (e.g., *"Location: {country}. Question: {question} Options: {options} Short Answer:"*); and a location-agnostic prompt, which follows the same template but does not specify the country in the prompt (e.g., *"Question: {question} Options: {options} Short Answer:"*). Additionally, due to the multilingual nature of CVQA, for each prompt, we evaluate using the English-only and local language question-option pairs. For the generative-based models, LLaVA, mBLIP and InstructBLIP, the image and the prompts are used as the input. The models then produce output probabilities and we treat the highest probability for the options (A,B,C,D) as the prediction (following MMLU [16]). On the other hand, for embedding-based models like CLIP and M-CLIP, we use the embedding-level similarity between the image and the combination of question and each answer candidate texts (*Question+Option-1,...,Question+Option-4*) to select the one with the highest similarity as the correct answer. We use accuracy to measure the performance, following the existing multiple-choice VQA tasks [2, 55].

# 4 Results

In this section, we discuss the performance of existing MLLMs on the CVQA benchmark.

Table 3: Average performance of MLLMs on our CVQA dataset with English prompts (EN) and local language prompts (LOC).

| LLaVA-1.5-7B | | M-CLIP | | CLIP | | mBLIP-mT0 | | mBLIP-BLOOMZ | | InstructBLIP | | Gemini-1.5-Flash | | GPT-4o | |
|------|------|------|------|------|------|------|------|------|------|------|------|------|------|------|------|
| EN | LOC | EN | LOC | EN | LOC | EN | LOC | EN | LOC | EN | LOC | EN | LOC | EN | LOC |
| 49.6 | 35.5 | 38.0 | 33.7 | 42.7 | 30.6 | 31.3 | 30.9 | 39.3 | 32.7 | 49.0 | 31.9 | 66.9 | 68.5 | 75.4 | 74.3 |

Table 4: LLaVA-1.5-7B and InstructBLIP results on various VQA datasets, where the results on the other datasets are taken from Liu et al. [26].

| Model | VQAv2 [13] | GQA [17] | VizWiz [15] | SciQA-IMG [28] | TextVQA [43] | CVQA (EN) | CVQA (LOC) |
|------|------|------|------|------|------|------|------|
| LLaVA-1.5-7B | 78.5 | 62.0 | 50.0 | 66.8 | 58.2 | 48.9 | 36.5 |
| InstructBLIP | - | 49.2 | 34.5 | 60.5 | 50.1 | 47.8 | 32.7 |

**Main Results** The overall performance on our CVQA dataset of various open and closed-source MLLMs are shown in Table 3. Among open models, LLaVA-1.5-7B achieves the best performance, but still significantly lagging behind closed models by more than 10%. However, Table 4 shows that LLaVA-1.5-7B indeed achieves better performance on other established English VQA benchmarks, highlighting that culturally-specific questions that we collect in CVQA are challenging even for the best-performing open model (LLaVA-1.5-7B). The performance is even worse when the question is asked in local languages, emphasizing the models' lower capability in handling non-English prompts.

The experimental results also highlight a substantial performance gap between open and closed-source MLLMs. Closed models like GPT-4o and Gemini-1.5-Flash demonstrate superior performance, with GPT-4o achieving the highest accuracy in both English (75.4%) and local language (74.3%) prompts. In contrast, open models like InstructBLIP and mBLIP-mT0 exhibit lower performance, particularly in local language prompts, indicating a need for more diverse training data and refined fine-tuning processes. While proprietary models show superior performance, it is hard to fully explain why, due to their closed nature. Additionally, their results are not reproducible. Therefore, we use open models in the rest of our experiments.

**Performance per Country-Language.** To see the capability of MLLMs in solving questions for each country and language, we report accuracy performance for Country-Language pairs in Figure 3. From this, we observe that all models struggle with questions in local languages, demonstrating the challenges for current MLLMs. In other words, across all models, their performance drops in local language questions compared to their performance in English questions. For instance, in the

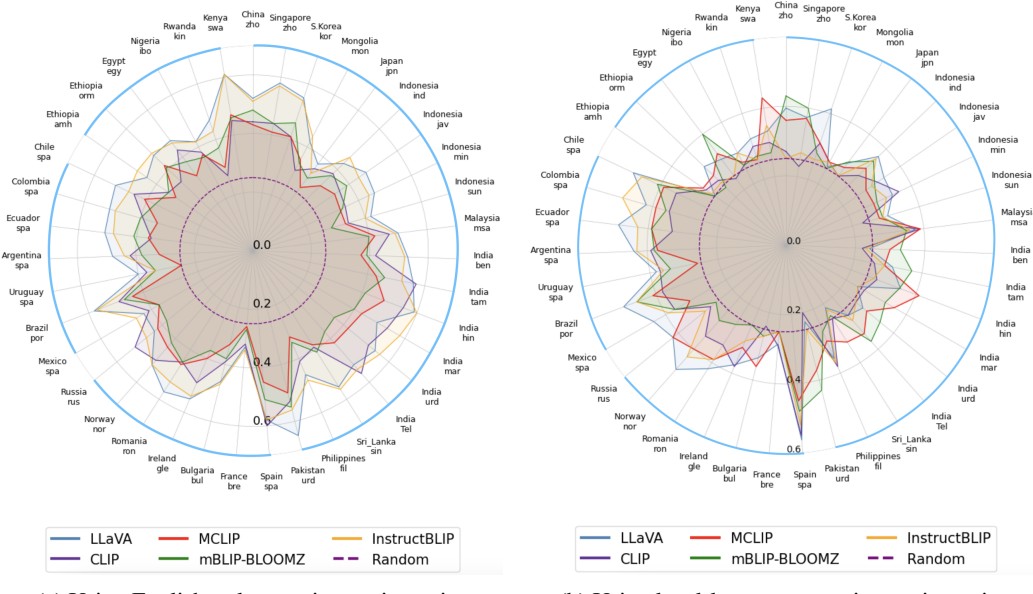

(a) Using English-only question-option pairs      (b) Using local-language question-option pairs

Figure 3: Model performance per Country-Language pair. The blue lines indicate separation by continent. All models show similar behaviour in the majority of cases, despite having different sizes.

case of Brazil-Portuguese, LLaVA-1.5-7B achieved a score of 60.73% for English and 51.16% for Portuguese. Moreover, in Mongolia-Mongolian, all models struggled, with LLaVA-1.5-7B reaching only 40% for English and 27.62% for Mongolian, suggesting challenges in less resource-rich language environments. It is worth noting that, these multilingual MLLMs do not originally support some of the languages, which also explains the significant performance drop for those languages. In contrast, in languages that are more frequently studied in NLP and have more abundant training resources, the performance gap between English and local languages, such as Spanish, tends to be smaller [3].

**Performance Across Categories.** We show the breakdown performances of models per category in Table 5, where the categories themselves are described in Section 2.1. Note that the category *People and Everyday Life* consistently achieves the highest accuracy across most models, with InstructBLIP obtaining 59.8% in English prompts. This can be possibly attributed to the extensive training data available for everyday human activity and interaction, which widely existed in many visual-related datasets. Conversely, the *Cooking & Food* and *Pop Culture* categories exhibit lower accuracy across models, especially in local language prompts. This demonstrates that the high diversity in food and pop culture across different cultures poses a great challenge for the generalization of MLLMs.

Table 5: Accuracy of models across categories. Per category, the best performing models on English (EN) and local language (LOC) question-option pairs are bolded and underlined, respectively.

| Categories | LLaVA-1.5-7B | | M-CLIP | | CLIP | | mBLIP-mT0 | | mBLIP-BLOOMZ | | InstructBLIP | |
|---|---|---|---|---|---|---|---|---|---|---|---|---|
| | EN | LOC | EN | LOC | EN | LOC | EN | LOC | EN | LOC | EN | LOC |
| Brands | **49.9** | 36.5 | 37.2 | 35.7 | 36.6 | 29.7 | 33.7 | 30.8 | 40.5 | 35.1 | 48.4 | 32.6 |
| Food | **45.4** | 31.9 | 34.5 | 29.1 | 39.2 | 30.4 | 28.1 | 27.6 | 37.7 | 29.8 | 44.4 | 30.6 |
| Geography | **47.1** | 38.2 | 37.1 | 34.2 | 41.8 | 31.9 | 30.6 | 31.6 | 35.0 | 32.3 | 45.3 | 33.2 |
| Objects | 51.8 | 33.0 | 39.4 | 34.5 | 39.7 | 25.4 | 34.3 | 33.0 | 43.1 | 34.0 | **52.3** | 29.1 |
| People | 58.9 | 38.1 | 45.0 | 37.8 | 46.8 | 30.9 | 35.3 | 34.7 | 46.3 | 36.7 | **59.8** | 34.0 |
| Plants & Animals | **55.7** | 37.5 | 43.7 | 32.0 | 48.0 | 27.2 | 35.2 | 35.5 | 46.0 | 36.0 | 55.4 | 35.1 |
| Pop Culture | 44.5 | 36.3 | 33.7 | 31.5 | **46.1** | 36.3 | 28.8 | 29.9 | 35.7 | 30.7 | 45.1 | 34.6 |
| Sports | **50.7** | 39.1 | 39.3 | 33.3 | 43.5 | 32.4 | 32.6 | 31.4 | 40.1 | 34.9 | 50.5 | 34.7 |
| Tradition | **50.4** | 35.8 | 37.0 | 35.2 | 41.9 | 32.2 | 31.6 | 31.5 | 39.0 | 32.2 | 47.9 | 30.8 |
| Vehicles | 50.6 | 41.4 | 39.5 | 41.1 | 44.6 | 30.5 | 35.6 | 33.9 | 42.0 | 34.0 | **55.0** | 33.0 |

**Impact of External Image Source.** The performance of various models on self-made versus web images is shown in Table 6. One of the interesting findings is the performance variability across

Table 6: Accuracy of different models divided by image source

| Image Source | LLaVA-1.5-7B | | M-CLIP | | CLIP | | mBLIP-mT0 | | mBLIP-BLOOMZ | | InstructBLIP | |
|---|---|---|---|---|---|---|---|---|---|---|---|---|
| | EN | LOC | EN | LOC | EN | LOC | EN | LOC | EN | LOC | EN | LOC |
| Self-made Image | 48.8 | 34.2 | 38.1 | 34.3 | 41.2 | 30.1 | 31.2 | 31.5 | 40.1 | 33.4 | 48.3 | 31.5 |
| Web Image | 49.7 | 37.4 | 37.4 | 33.3 | 43.1 | 31.8 | 31.9 | 31.2 | 38.7 | 32.3 | 49.1 | 33.1 |

Table 7: Location-aware and location-agnostic results

| Prompt type | LLaVA-1.5-7B | | M-CLIP | | CLIP | | mBLIP-mT0 | | mBLIP-BLOOMZ | | InstructBLIP | |
|---|---|---|---|---|---|---|---|---|---|---|---|---|
| | EN | LOC | EN | LOC | EN | LOC | EN | LOC | EN | LOC | EN | LOC |
| Location-aware | 49.6 | 35.5 | 38.0 | 33.7 | 42.7 | 30.6 | 31.3 | 30.9 | 39.3 | 32.7 | 49.0 | 31.9 |
| Location-agnostic | 48.3 | 34.7 | 38.1 | 33.8 | 43.8 | 30.8 | 34.1 | 31.8 | 39.8 | 33.6 | 48.7 | 31.1 |

image sources for different models. For self-made images, the accuracy of some models such as LLaVA-1.5-7B and CLIP tends to be lower compared to web images. For instance, LLaVA-1.5-7B achieves a 48.8% accuracy in English prompts on self-made images but slightly higher at 49.7% on web images. CLIP shows an accuracy of 43.1% in English prompts on web images compared to 41.2% on self-made images. While this trend is not consistent across the other models, the results still indicate that web images might be more representative of the data these models (such as LLaVA-1.5-7B and CLIP) were trained on, leading to better performance.

**Location-Aware vs Location-Agnostic Prompt.** The performance of the various models on location-aware versus location-agnostic prompts is shown in Table 7. While the inclusion of location information has a varied impact on different models, the overall difference between both prompt options is marginal, suggesting no significant effect of including location information on MLLMs.

**Performance without Multiple Choice Options.** Most of the evaluations we conduct on CVQA are under a multiple-choice setting. However, the multiple-choice setting is often brittle towards option ordering [38, 54], and not very natural with respect to real-world scenarios [30]. In this paragraph, we explore the model's performance on CVQA in an open-ended QA setting. To evaluate in this setting, we prompt the models without giving them the options (e.g., "In which city is this monument located?"). Then, the answer is selected by choosing the model's highest probability of generating the full answer phrase of one of the options [11] (e.g., Jakarta, Bandung, Bali, Surabaya). This way, it is robust towards ordering unlike predicting the answer letter (e.g., A), while also not giving the model multiple-choice options that can be indirectly used for deductive reasoning. Our result shows that LLaVA-1.5-7B achieved a noticeable performance drop when prompted without multiple choice, from 49.6% to just 30% average performance. This notes that in a more practical scenario, these models might be even more unreliable in cultural understanding.

## 5 Limitations

Our new benchmark dataset represents a diverse worldview through the inclusion of different languages and regions not covered in previous datasets. But we acknowledge that even CVQA is not comprehensive, as it covers only a fraction of the world's languages and regions. CVQA also lacks an English-centric baseline, which could arguably provide an interesting comparison with the rest of the regions. Additionally, our data scale prevents using CVQA to train new models, limiting its use for benchmarking purposes only.

We note that each region has different characteristics of questions and difficulty—some regions are more likely to provide simpler, identity "what is" questions, whereas other regions might use questions that require deeper cultural knowledge. Therefore, comparing performance across languages/countries might not always be fair.

Culture is hard to define, and our CVQA ultimately serves only as a proxy to benchmark the model's understanding of culture through local common knowledge. However, this by no means captures all cultural nuances [1]. Additionally, our location granularity captures country-level cultural knowledge. However, it might be interesting to capture cultural awareness at a more granular level, such as city-level, since each city might have variations in cultural common knowledge. Similarly, other demographic factors such as age might play a role in common knowledge.

In this section we discussed the following aspects: 1) the fact that this dataset cannot be considered as a comprehensive representation of the world languages and regions; 2) the different levels of question complexity; 3) a bounded definition of culture. While these limitations might be relevant, we consider them as plausible lines for future work and outside the scope of this initial effort.

# 6 Related Work

Substantial progress has been made in recent years on both datasets and methodologies for VQA [42, 23]. Since the introduction of early open-ended VQA datasets [2, 13], various formats like multiple-choice [49, 28], span extraction [34], and free-text generation [27] have been developed. Among these, multiple-choice datasets [28, 29, 52] are the most commonly used, likely due to their simplicity in evaluation and comparison. The development of these datasets has significantly accelerated research progress, serving as both training data and testbeds, especially the recently introduced ScienceQA [28] and MathVista [29] designed for evaluating MLLMs. The evolution of VQA methodologies has been revolutionary, transitioning from statistical machine learning [31] to neural-based methods [32, 17, 40], and advanced MLLMs [27, 36, 45] trained on massive multimodal data. Early VQA systems often required supervised learning and were limited to specific domains, but recent models like CLIP [40], LLaVA [27], and GPT-4V [36] are capable of zero-shot or few-shot learning, demonstrating strong performance. Despite this progress, significant limitations remain. Most VQA datasets focus primarily on English and a few major world languages [28, 29, 51], leading to language bias and under-representation of many languages and cultures. Additionally, the images in these datasets predominantly reflect Western scenes and styles, lacking the diversity needed to represent real-world scenarios across different cultures [7].

Some efforts have been made to create multilingual VQA datasets, such as FM-IQA [10], MCVQA [14], xGQA [39], MaXM [6], MTVQA [44], and MaRVL [25]. However, these datasets are still limited in terms of the number of languages and the cultural diversity of the images and questions, or being a translation of existing English data. On the other hand, there have been initiatives to create culturally-diverse datasets and benchmarks under text-only modality [35, 19, 48, 18, 9, 47, 21]. Our proposed benchmark aims to fill the gap that covers both textual and visual modality by creating a large-scale, culturally-and-linguistically diverse dataset that will enable the development of more inclusive and robust VQA models.

# 7 Conclusion

We proposed CVQA, a novel, human-written visual QA benchmark dataset that captures cultural nuances across a diverse set of languages and locations. CVQA encompasses 10 question categories, with each question written in both English and the native language. This allowed us to benchmark both multilingual visual models and English-only models. We provided insights into our dataset's question types and commonly used terms for each category.

We then performed benchmarks on various visual models, including both multilingual and English-only models. Our benchmark demonstrated that CVQA presented challenges for open-source models. These models generally performed worse when queried in local languages compared to English, indicating poorer performance in handling multilingual queries. The performance is also considerably lower when we do not provide the multiple choice setting, which is a more realistic use case for this technology. We hope that publishing CVQA encourages the AI community to pay more attention to non-English-centric models and benchmarking, thereby advancing progress in multilingual, multimodal research.

## Acknowledgments

We thank Ananjay Goel, Aditya Rachman Putra, Radityo Eko Prasojo, Amr Keleg, and Mostafa Awad for the contributions in creating the dataset. Tiago Torrent was partially funded by CNPq Research Productivity Grant number 315749/2021-0. Luis Fernando D'Haro, Mario Rodriguez-Cantelar and Marcos Estecha-Garitagoitia were supported by the European Commission through Project ASTOUND (101071191 — HORIZON-EIC-2021- PATHFINDERCHALLENGES-01), and by project BEWORD (PID2021-126061OB-C43) funded by MCIN/AEI/10.13039/501100011033

and, as appropriate, by "ERDF A way of making Europe", by the "European Union". We also thank the anonymous reviewers for their valuable feedback and suggestions that helped improve this paper.

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

## A    Annotation Guidelines

# Multilingual Multimodal Visual Question Answering

# Benchmark: Annotation Guidelines

### Introduction

This document provides guidelines for annotating images and corresponding questions and answers in multiple languages to create a culturally diverse and linguistically comprehensive multimodal QA benchmark.

### Objective

To build a benchmark that represents a wide range of cultures and languages, to measure potential bias in visual AI models.

### Guidelines for Contributors

Each region and language (eg. Ecuador-Spanish) will be represented by **at most 3 annotators, in which 1 will be the team lead**. Each person is expected to provide at least 100 visual questions to be considered as a co-author. The team lead will still have to provide questions, the only difference is that the team lead is responsible to find and to organize more annotators and will manage to contact and brief that annotator, if needed.

**Image Selection:**

- ○ Contribute images that represent diverse cultural aspects that represent the specific cultural background you're contributing to. The image must fall into one of the categories below. **Pick one of the most relevant category (more later)**:

  Image Category *

  ☐ Vehicles and Transportation          ☐ Brands, products, and companies

  ☐ Objects, materials, clothing          ☐ Sports and recreation

  ☐ Cooking and food                      ☐ Traditions, art, and history

  ☐ Geography, buildings, and landmarks   ☐ People and everyday life

  ☐ Plants and animal                     ☐ Public Figure and pop culture

  ☐ Other

- ○ Images should be relevant to your culture/country.
- ○ Ensure that **images are relevant to the questions being posed.** In other words, the image **is needed** to answer the question.
- ○ If the image contains the answer's text, you can blur/crop the image so that the image does not contain the answer.

- ○ Image source:
    - ■ 1. Self/personal picture **(highly preferable).** You may ask your family/friend to donate their photos, if possible.
    - ■ 2. We also accept external images from:
        - ● Flickr: https://www.flickr.com/explore (**please make sure the associated license to the image is Creative Commons**), this can be selected at the the top left of Flickr ("Any License").
        - ● WikimediaCommons: https://commons.wikimedia.org/wiki/Main_Page (**here you do not need to select any license for the images)**,
        - ● Unsplash: https://unsplash.com/ (**please make sure to search the image first and they select the license: Free**). More details (Tutorial) at the end of this document.
        - ● Dollar Street: https://www.gapminder.org/dollar-street (**here you do not need to select any license for the images),** this webpage has images only from some countries, please make sure to select your country to find images if applicable.

            More detailed instructions for each web page are shown at the end of this document.

    - ■ If you use an external image, you'll need to put the url of the original image.
- ○ The image must be reasonable quality (not pixelated or blurry, can be understandable). You can upload images of any ratio as long as it is not too tall or wide (e.g.: don't submit panorama pictures).
- ○ Do not show personally identifiable information (PII) such as faces, car plates, house addresses. Faces of public figures or fictional characters are ok. Also, **please be sure to blur text in the image that will leak the answer**. "PicdeFacer" can be used for blurring: https://picdefacer.com/en/. Tutorial on using PicdeFacer is shown at the end of this document.

**Question and Answer Creation:**

After finding the image, you must now formulate 1-3 questions + answers from that image. Specifically:

- ○ The question must be answerable **only by looking at the image.**
- ○ Ensure that the questions are culturally relevant and specific to the image content.
- ○ Provide answers that are concise, accurate, and directly related to the question.
- ○ You will also need to provide 1 correct option and 3 other incorrect options (distractors). For the distractors, choose options that are relevant, not obvious wrong answers.

- **The question must be answerable even without the multiple-choice.**
  Example of the invalid question: ("What song is not performed by this musician" – not answerable if you don't know the choices)
- Make sure the questions are **written fluently in both the local language and English.** Use a grammar checker if needed i.e. if you are not fluent in English.
- Be mindful of cultural sensitivities and avoid stereotyping or misrepresenting cultural aspects.
- Ensure there are **variations on your question**. Identity questions are fine, eg "What is this", or "where is this". But additionally adding more complex/difficult questions would be great. For example, multi-hop reasoning, counting, referencing, or questions that require local commonsense knowledge to be answered.

## Category Definition

When selecting a category, pick one of the most relevant. Please follow the guideline:

- **Vehicles and Transportation:** Local public transport, local vehicles.

- **Objects, Materials, Clothing:** Questions about local/traditional clothes. Unique/local tools or items.

- **Cooking and Food:** Local dishes and food/drink. This category includes native fruits in the context of the image if that fruit is served as a food/drink.

- **Geography, Buildings, Landmarks:** Popular/common landmarks, local architecture/buildings. Local monuments.

- **Plants and Animals:** Plants and animals commonly found in the region.

- **Brands, Products, and Companies:** Questions about understanding local yet popular brands or companies. Even if the brand is about food/transportation, if the main focus of the question is the brand recognition itself, then it should be under this category.

- **Sports & Recreation:** Local sports and fun activities. Focuses on the activity itself rather than the location (in that case, it goes to the 'landmark' category).

- **Tradition, Art, History:** Local ceremonies/festivals/events, local dance/music, folklores. Historical artifacts.

- **People & Everyday Life:** Focuses on the people themselves: i.e., common habits/customs, common occupations and jobs, routine religious activities, everyday activities/routines.

- **Public Figures & Pop Culture:** Questions on the understanding of common public figures (e.g., politicians, artists, musicians, etc.). Common pop culture such as movies and games.
If the category is still ambiguous to you, pick the one you think is the most appropriate.

## Examples

**Examples that can be improved**

| | |
|---|---|
| | **Make sure the image is needed to respond the question, example:** 

 1) ¿En qué mes se celebra esta fiesta? 
 (In which month is this celebration held?) 
 Correct 
 2) ¿En qué mes se celebra la fiesta de la "Mama Negra"? 
 (In which month is the celebration of the "Mama Negra" held?) 
 Wrong–As this question can be answered without looking at the image. |
| | **Make sure the question is not ambiguous:** 
 1) Where is this monument located? 
 Wrong–Not specific, the answer could be a city, country, province,etc. 
 2) In which **city** is this monument located? 
 Correct–specifically asking about the city |
| | **Make sure the question is not too vague:** 
 1) What is this? 
 Question wording can be more specific 
 2) What is the name of this vehicle? 
 Correct–specifically asking about the vehicle name. |

**Acceptable examples**

| | |
|---|---|
| 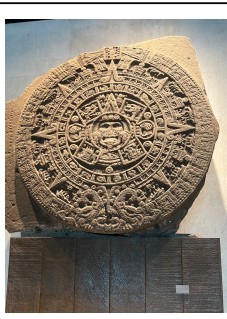 | **Category:Tradition / Art / History – Spanish/Mexico** 

 ¿Qué se muestra en la imagen?  (What is shown in the image?) 
 **A. el calendario azteca/ piedra del sol (the aztec calendar/ aztec sun stone)** 
 B. una serpiente azteca (an aztec serpent) 
 C. coatlicue (coatlicue) 
 D. tláloc (tlaloc) 

 ¿En dónde se exhibe esta pieza?  (Where is this piece exhibited?) 
 **A. En el museo nacional de antropología (In the National Museum of Atroplogy)** 
 B. en el castillo de Chapultepec (In the Chapultepec Castle) 
 C. En el zócalo de la ciudad de Mexico (In the Mexico City zocalo) 
 D. En Teotihuacan (In Teotihuacan) |

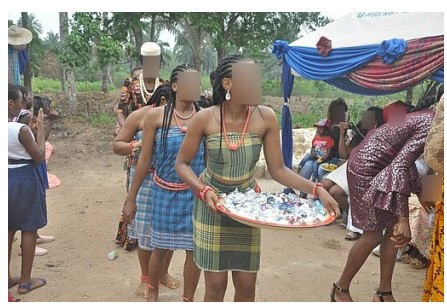

**Category: Tradition/ Art / History – Igbo/Nigeria**

Kedụ mmemme ndị a na-eme?
(Which ceremony are they doing?)

- **A. Ị gba nkwụ (Traditional marriage)**
- B. Ncheta Ọmụmụ (Birthday)
- C. Emume cheiftaincy (Chieftaincy ceremony)
- D. Emume iri ji ọhụrụ (New yam festival)

Kedụ ebe a na-eme mmemme a?
(Where is this ceremony held?)

- **A. Ụlọ nna nwunye (The home of the bride's father)**
- B. Ahia (The market)
- C. Ụlọ nna di (The home of the groom's father)
- D. Ụlọ nsọ (The church)

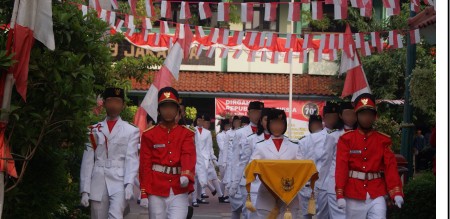

**Category: Tradition/ Art / History – Indonesian/Indonesia**

Pada tahun berapakah foto ini diambil?
(In what year is this photo taken?)

- **A. 2015 (2015)**
- B. 2020 (2020)
- C. 2023 (2023)
- D. 2010 (2010)

Apa nama pasukan yang ada di foto ini?
(What is the name of the squad in this photo?)

- **A. Paskibraka (Paskibraka)**
- B. Brimob (Brimob)
- C. TNI (TNI)
- D. ABRI (ABRI)

Apa tugas utama pasukan ini?
(What is the main purpose of this squad?)

- **A. Mengibarkan bendera (Hoisting the flag)**
- B. Mengawal presiden (Escorting president)
- C. Menjaga keamanan (Maintaining security)
- D. Mengiringi pengantin (Accompanying the bride and groom)

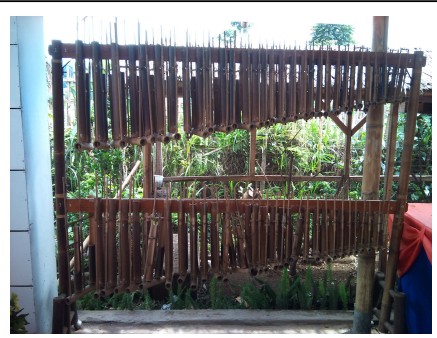

**Category: Tradition, Art, History – Sundanese/Indonesia**

Naon kagunaan ieu hiji alat?
(What is the use of this tool?)

**A. Alat musik (Musical instrument)**
B. Alat pertahanan diri (Self defence tool)
C. Jemuran (Clothes drying equipment)
D. Alat masak (Cooking tool)
–

Ieu hiji alat teh asalna ti propinsi mana di Indonesia?
(This tool comes from which province in Indonesia?)

**A. Jawa Barat (West Java)**
B. Bali (Bali)

| | |
|---|---|
| | C. Bengkulu (Bengkulu)
D. Sumatra Barat (West Sumatra) |
| 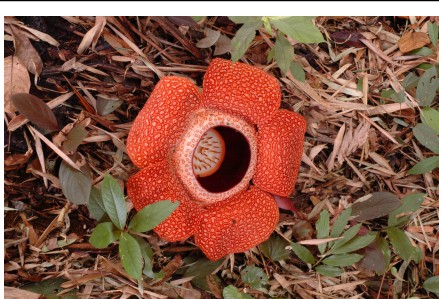 | **Category: Plants and animals – Malay/Malaysia**

Apakah nama bunga dalam gambar ini?
(What is the name of the flower in this picture?)

    **A.  Pakma (Rafflesia)**
    B.  Bunga raya (Hibiscus)
    C.  Anggerik (Orchid)
    D.  Bunga kertas (Bougainvillea)

Di rantau Asia manakah bunga itu boleh ditemui?
(In which region of Asia can the flower be found?)

    **A.  Asia Tenggara (Southeast Asia)**
    B.  Asia Timur (East Asia)
    C.  Asia Selatan (South Asia)
    D.  Asia Tengah (Central Asia) |
| 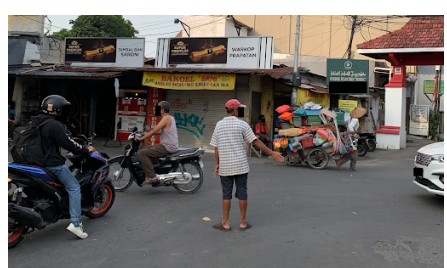 | **Category: People and everyday life - Javanese/Indonesia**

Opo arane wong seng nang tengah embong iki?
(What is the term for the man in the middle of the road?)

    **A.  Polisi cepek (Polisi cepek)**
    B.  Tukang parkir (Parking assistance man)
    C.  Mlijo (Grocery man)
    D.  Tukang becak (Pedicap man)

Opo seng dilakukno wong seng nang tengah dalan iku?
(What does the man in the middle of the road do?)

    **A.  Ngatur prapatan (Managing the intersection)**
    B.  Njaluk donasi (Asking for donations)
    C.  Ngawasi pelanggaaran lalu lintas (Looking out for traffic violations)
    D.  Nunjukno arah (Showing directions) |
| 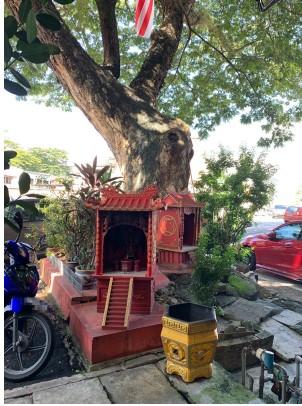 | **Category: People and everyday life – Malay/Malaysian**

Roh manakah yang disembah dengan altar ini?
(Which deity is worshiped on this altar?)

    **A.  Datuk Gong (Na Tuk Kong)**
    B.  Buddha (Buddha)
    C.  Brahma (Brahma)
    D.  Vishnu (Vishnu)

Apakah agama yang diamalkan oleh pengguna altar ini?
(What religion do the users of these altars practice?)

    **A.  Taoism (Taoisme)**
    B.  Buddha (Buddhisme)
    C.  Islam (Islam)
    D.  Hindu (Hinduisme) |

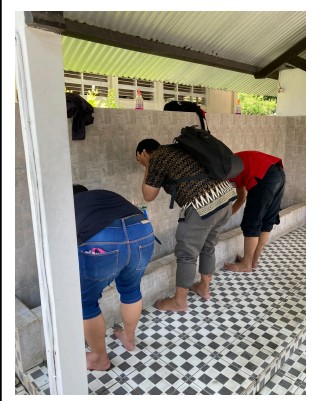

**Category: People and Everyday Life – Indonesian/Indonesia**

Apa yang orang-orang ini lakukan?
(What are these people doing?)

**A. Berwudhu (Performing ablution)**
B. Mandi (Taking a bath)
C. Yoga (Yoga)
D. Beribadah (Praying)

Dimana biasanya orang-orang melakukan aktivitas di foto ini?
(Where do people usually do the activity in this photo?)

**A. Masjid (Mosque)**
B. Gereja (Church)
C. Pemandian umum (Public bath)
D. Gym (Gym)

---

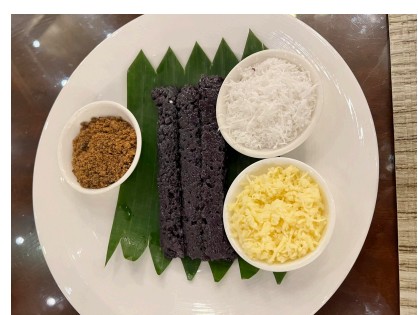

**Category: Cooking and Food – Tagalog/Philippines**

Anong tawag sa kakanin na ito?
(What is the name of this rice cake?)

    **A. Puto Bumbong (Puto Bumbong)**
    B. Suman (Suman)
    C. Kutsinta (Kutsinta)
    D. Sapin-Sapin (Sapin-Sapin)

Tuwing kailan ito madalas tinitinda sa Pilipinas?
(When is this food usually sold in the Philippines?)

    **A. Christmas Season (Christmas Season)**
    B. Independence Day (Independence Day)
    C. Labor Day (Labor Day)
    D. National Heroes Day (National Heroes Day)

Ano tawag dun sa brown?
(What do you call the brown object?)

    **A. Muscovado (Muscovado)**
    B. Latik (Toasted coconut)
    C. Chocolate (Chocolate)
    D. Caramel (Caramel)

| | |
|---|---|
| 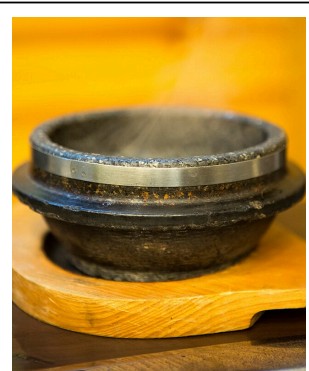 | **Category: Object, Clothing, and Material – Korean/South Korea**

이런 종류의 요리에 사용되는 그릇을 무엇이라고 부르나요?
(What is this type of bowl called in cooking?)

    **A. 돌솥 (Dolsot)**
    B. 복주머니 (Bokjumeoni)
    C. 냄비 (Pot)
    D. 팬 (Pan)

그릇의 재질은 무엇인가요?
(What is the material of the bowl?)

    **A. 돌 (Stone)**
    B. 도자기 (Ceramic)
    C. 유리 (Glass)
    D. 스테인리스 스틸 (Stainless Steel) |
| 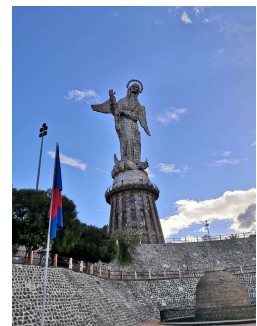 | **Category: Landmark and building - Spanish/Ecuador**

¿Cómo se llama este monumento ubicado en Quito?
(What is the name of this monument located in Quito?)

**A. Virgen de El Panecillo (The Virgin of El Panecillo)**
B. Manto de María (Manto de María)
C. Mitad del mundo (Middle of the world)
D. Cristo de la concordia (Christ of peace) |
| 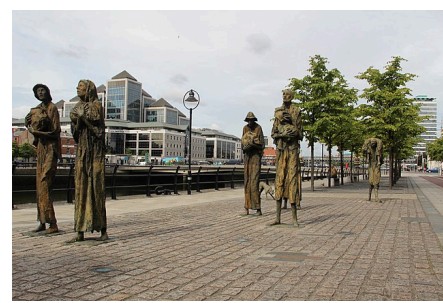 | **Category: Landmark and building - Irish/Ireland**

Cén cathair ina bhfuil na dealbha seo?
(In which city are these statues?)

    **A. Cathair Bhaile Átha Cliath (Dublin City)**
    B. Páras (Paris)
    C. Cathair Corcaigh (Cork City)
    D. Beirlín (Berlin)

Cén eachtra stairiúil atá léirithe sna dealbha seo?
(What historical event is depicted in these statues?)

    **A. An Ghorta Mhór (The Great Famine)**
    B. Éirí Amach 1916 (The 1916 Rising)
    C. Teitheadh na n-Iarlaí (The flight of the Earls)
    D. Cogadh 1835 (The 1835 war)

Cén abhainn atá le taobh na ndealbh seo?
(What river is beside these statues?)

    **A. An Life (The Liffey)**
    B. An tSionann (The Shannon)
    C. Abhainn an Rí (King's River)
    D. An Thames (The Thames) |

## B   Annotation Platform

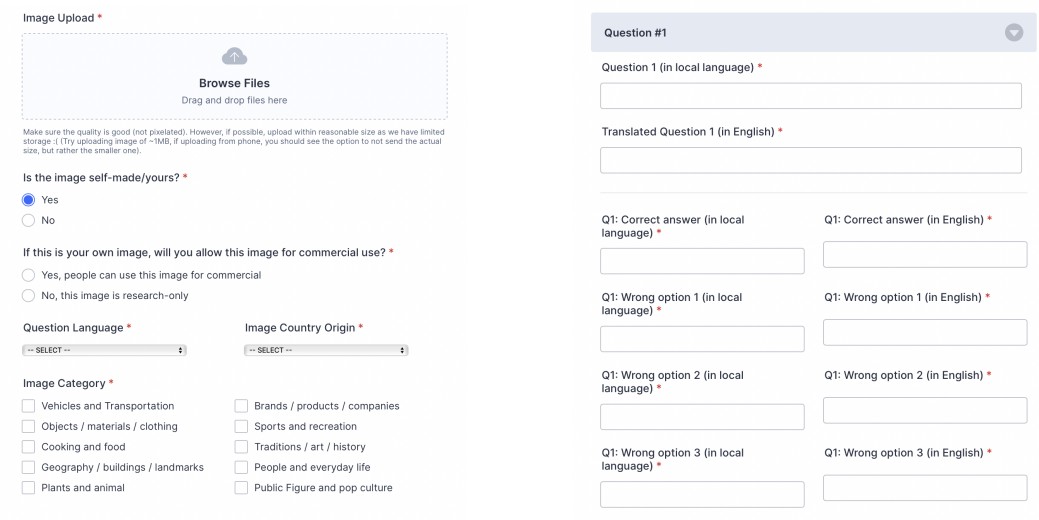

Figure 4: Annotation interface for inputting image and questions

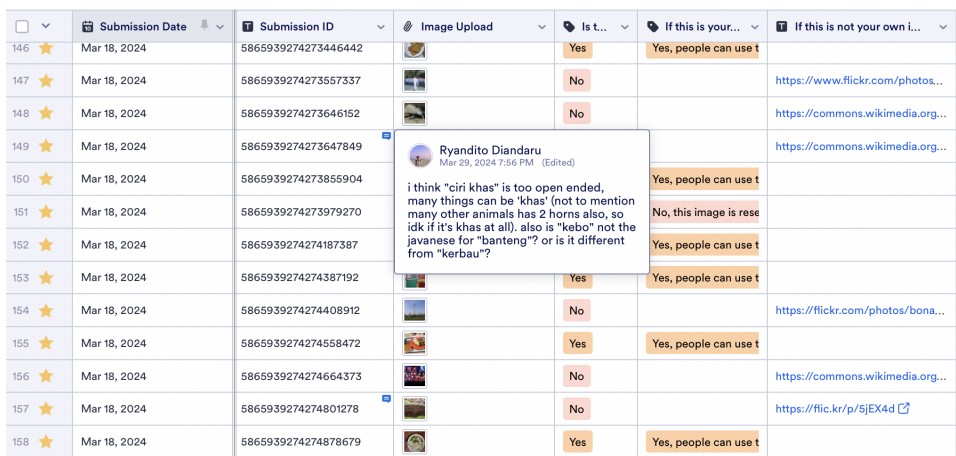

Figure 5: Annotation interface for validation. Contributors can comment, edit, and star the entries

We use JotForm as our annotation platform. For question entry, contributors can upload and write questions in both languages in the form. The interface can be seen in Figure 4. During validation, contributors can see all the data submitted by other contributors (Figure 5). They can select the entry to see detailed preview of the entry (Figure 6). They can then either edit the data directly, provide comments, or confirm the data by starring the entry.

## C   Most-Frequent Words in the Questions

Figure 7 shows word clouds for the most frequent words in CVQA per category. We exclude stopwords as well as 'picture', 'photo', and 'image' from the list, since most questions contain these words. In this VQA context, we can treat them as stopwords.

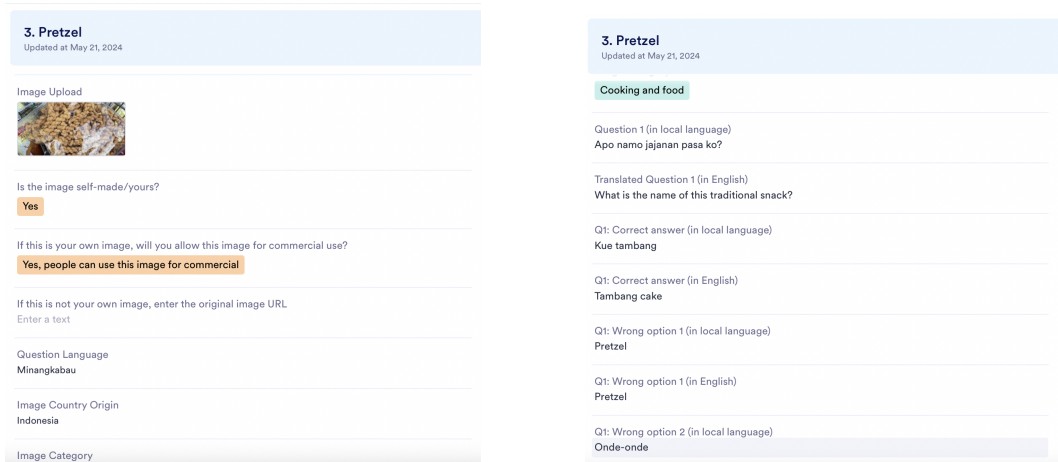

Figure 6: During validation, contributors can preview the submission from other contributors

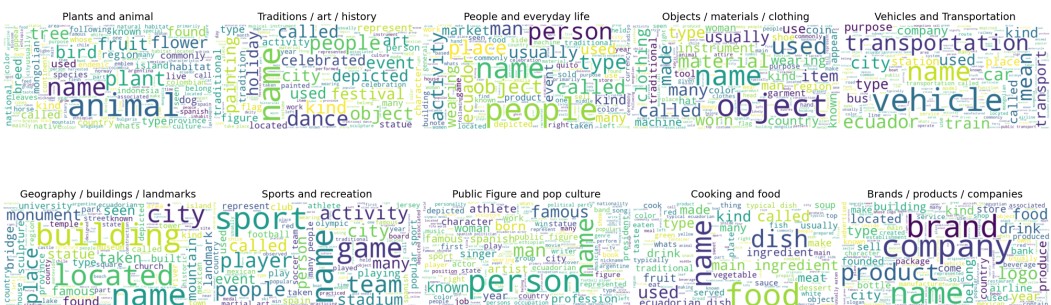

Figure 7: Word Cloud in CVQA per category

# D  CVQA Annotator Demographic

Figure 8 illustrates the demographic statistics of the annotators, based on an anonymous questionnaire we provided. At the time of writing, we have information for 36 out of 76 annotators. As such, this breakdown is a rough representation of the annotation group.

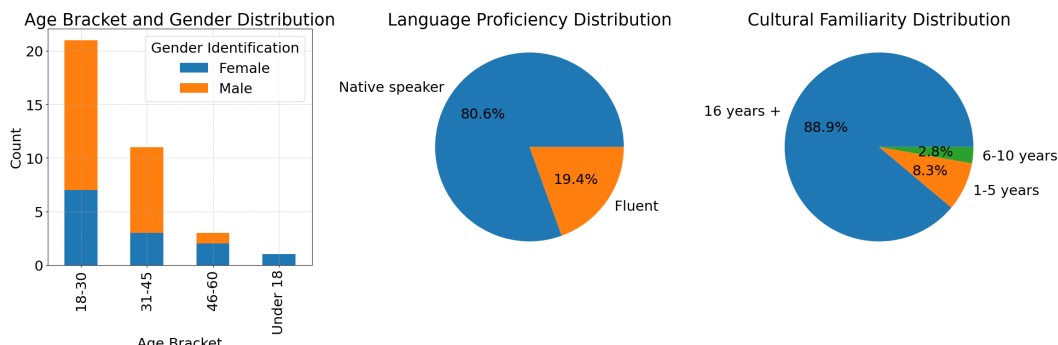

Figure 8: Annotator demographic statistics

# E  Country-Language Pairs and Scripts

In Table 8, we provide information on the script used in each Country-Language pair.

| Country | Language | Script |
|---|---|---|
| *Africa* | | |
| Egypt | Egyptian Arabic | Arabic |
| Ethiopia | Amharic | Amharic |
| Ethiopia | Oromo | Latin |
| Kenya | Swahili | Latin |
| Nigeria | Igbo | Latin |
| Rwanda | Kinyarwanda | Latin |
| *Asia* | | |
| China | Chinese | Chinese |
| India | Bengali | Bengali |
| India | Hindi | Devanagari |
| India | Marathi | Devanagari |
| India | Tamil | Tamil |
| India | Telugu | Telugu |
| India | Urdu | Perso-Arabic |
| Indonesia | Indonesian | Latin |
| Indonesia | Javanese | Latin |
| Indonesia | Minangkabau | Latin |
| Indonesia | Sundanese | Latin |
| Japan | Japanese | Japanese |
| South Korea | Korean | Hangul |
| Malaysia | Malay | Latin |
| Mongolia | Mongolian | Cyrillic |
| Pakistan | Urdu | Perso-Arabic |
| Philippines | Filipino | Latin |
| Singapore | Chinese | Chinese |
| Sri Lanka | Sinhala | Sinhalese |
| *Europe* | | |
| Bulgaria | Bulgarian | Cyrillic |
| France | Breton | Latin |
| Ireland | Irish | Latin |
| Norway | Norwegian | Latin |
| Romania | Romanian | Latin |
| Russia | Russian | Cyrillic |
| Spain | Spanish | Latin |
| *Latin America* | | |
| Argentina | Spanish | Latin |
| Brazil | Portuguese | Latin |
| Chile | Spanish | Latin |
| Colombia | Spanish | Latin |
| Ecuador | Spanish | Latin |
| Mexico | Spanish | Latin |
| Uruguay | Spanish | Latin |

Table 8: The list of Country-Language pairs covered in CVQA and their corresponding scripts.

# F    Affiliation Lists

Table 9 lists the authors and their respective affiliations.

Table 9: Author affiliations

| Author | Affiliation | Author | Affiliation | Author | Affiliation |
|---|---|---|---|---|---|
| David Romero | MBZUAI | Chenyang Lyu | MBZUAI | Haryo Akbarianto Wibowo | MBZUAI |
| Teresa Lynn | MBZUAI | Injy Hamed | MBZUAI | Aditya Nanda Kishore | IIT Madras |
| Aishik Mandal | TU Darmstadt | Alina Dragonetti | Universidad de la República | Artem Abzaliev | University of Michigan |
| Atnafu Lambebo Tonja | Independent Researcher | Bontu Fufa Balcha | Independent Researcher | Chenxi Whitehouse | University of Cambridge |
| Christian Salamea | Universidad Politécnica Salesiana | Dan John Velasco | Samsung Research Philippines | David Ifeoluwa Adelani | Independent Researcher |
| David Le Meur | Bretagne numérique | Emilio Villa-Cueva | MBZUAI | Fajri Koto | MBZUAI |
| Fauzan Farooqui | Independent Researcher | Frederico Belcavello | Federal University of Juiz de Fora | Ganzorig Batnasan | United Arab Emirates University / MBZUAI |
| Gisela Vallejo | The University of Melbourne | Grainne Caulfield | Dublin City University | Guido Ivetta | Universidad Nacional de Córdoba |
| Haiyue Song | NICT | Henok Biadglign Ademtew | EAII | Hernán Maina | Universidad Nacional de Córdoba/CONICET |
| Holy Lovenia | AI Singapore | Israel Abebe Azime | Saarland University | Jan Christian Blaise Cruz | Samsung Research Philippines |
| Jay Gala | MBZUAI | Jesus-German Ortiz-Barajas | MBZUAI | Jiahui Geng | MBZUAI |
| Jinheon Baek | KAIST | Jocelyn Dunstan Escudero | Pontificia Universidad Católica de Chile | Kumaranage Ravindu Yasas Nagasinghe | MBZUAI |
| Laura Alonso Alemany | Universidad Nacional de Córdoba | Luciana Benotti | Universidad Nacional de Córdoba/CONICET | Luis Fernando D'Haro | Universidad Politecnica de Madrid |
| Marcelo Viridiano | Federal University of Juiz de Fora | Marcos Estecha-Garitagoitia | Universidad Politécnica de Madrid | Maria Camila Buitrago Cabrera | University of Stuttgart |
| Mario Rodríguez-Cantelar | Universidad Politécnica de Madrid | Mélanie Jouitteau | IKER, CNRS | Mihail Mihaylov | MBZUAI |
| Mohamed Fazli Mohamed Imam | MBZUAI | Muhammad Farid Adilazuarda | MBZUAI | Munkh-Erdene Otgonbold | United Arab Emirates University |

| Author | Affiliation | Author | Affiliation | Author | Affiliation |
|---|---|---|---|---|---|
| Munkhjargal Gochoo | United Arab Emirates University | Naome A. Etori | Independent Researcher | Olivier NIYOMUGISHA | Independent Researcher |
| Paula Mónica Silva | Millenium Institute Foundational Reseach on Data | Pranjal Chitale | Independent Researcher | Raj Dabre | IIT Madras |
| Rendi Chevi | MBZUAI | Ruochen Zhang | Brown University | Ryandito Diandaru | ITB |
| Samuel Cahyawijaya | HKUST | Santiago Góngora | Universidad de la República | Soyeong Jeong | KAIST |
| Sukannya Purkayastha | TU Darmstadt | Tatsuki Kuribayashi | MBZUAI | Thanmay Jayakumar | IIT Madras |
| Tiago Timponi Torrent | Federal University of Juiz de Fora, CNPq | Toqeer Ehsan | MBZUAI | Vladimir Araujo | KU Leuven |
| Yova Kementchedjhieva | MBZUAI | Zara Burzo | Skyline Highschool | Zheng Wei Lim | The University of Melbourne |
| Zheng-Xin Yong | Brown University | Oana Ignat | University of Michigan | Joan Nwatu | University of Michigan |
| Rada Mihalcea | University of Michigan | Thamar Solorio | MBZUAI | Alham Fikri Aji | MBZUAI |

