# CVQA: Culturally-diverse Multilingual Visual Question Answering Benchmark Supplementary Material

**David Romero**[*♣], **Chenyang Lyu**[*♣], **Haryo Akbarianto Wibowo**[♣], **Teresa Lynn**, **Injy Hamed**,
**Aditya Nanda Kishore**, **Aishik Mandal**, **Alina Dragonetti**, **Artem Abzaliev**,
**Atnafu Lambebo Tonja**, **Bontu Fufa Balcha**, **Chenxi Whitehouse**, **Christian Salamea**,
**Dan John Velasco**, **David Ifeoluwa Adelani**, **David Le Meur**, **Emilio Villa-Cueva**,
**Fajri Koto**, **Fauzan Farooqui**, **Frederico Belcavello**, **Ganzorig Batnasan**, **Gisela Vallejo**,
**Grainne Caulfield**, **Guido Ivetta**, **Haiyue Song**, **Henok Biadglign Ademtew**, **Hernán Maina**,
**Holy Lovenia**, **Israel Abebe Azime**, **Jan Christian Blaise Cruz**, **Jay Gala**, **Jiahui Geng**,
**Jesus-German Ortiz-Barajas**, **Jinheon Baek**, **Jocelyn Dunstan**, **Laura Alonso Alemany**,
**Kumaranage Ravindu Yasas Nagasinghe**, **Luciana Benotti**, **Luis Fernando D'Haro**,
**Marcelo Viridiano**, **Marcos Estecha-Garitagoitia**, **Maria Camila Buitrago Cabrera**,
**Mario Rodríguez-Cantelar**, **Mélanie Jouitteau**, **Mihail Mihaylov**, **Naome Etori**,
**Mohamed Fazli Mohamed Imam**, **Muhammad Farid Adilazuarda**, **Munkhjargal Gochoo**,
**Munkh-Erdene Otgonbold**, **Olivier Niyomugisha**, **Paula Mónica Silva**, **Pranjal Chitale**,
**Raj Dabre**, **Rendi Chevi**, **Ruochen Zhang**, **Ryandito Diandaru**, **Samuel Cahyawijaya**,
**Santiago Góngora**, **Soyeong Jeong**, **Sukannya Purkayastha**, **Tatsuki Kuribayashi**,
**Teresa Clifford**, **Thanmay Jayakumar**, **Tiago Timponi Torrent**, **Toqeer Ehsan**,
**Vladimir Araujo**, **Yova Kementchedjhieva**, **Zara Burzo**, **Zheng Wei Lim**, **Zheng Xin Yong**,
**Oana Ignat**, **Joan Nwatu**, **Rada Mihalcea**, **Thamar Solorio**[♣], and **Alham Fikri Aji**[♣]

♣Core Authors (MBZUAI)
www.cvqa-benchmark.org

In this Supplementary Material, we present the following items:

1. A datasheet for the dataset documentation of CVQA (Section 1).

2. The data access and maintenance plan (Section 2).

3. The Statement of responsibility (Section 3).

4. The annotation guideline provided to CVQA annotators (Section 4).

5. Annotation Platform (Section 5).

6. Additional Statistics (Section 6)

## 1 Datasheet for CVQA

For documenting CVQA, we use the datasheet for datasets introduced by Gebru et al. [1], which specify the motivation, composition, collection process, preprocessing, uses and distribution of a dataset. We follow and provide this datasheet for CVQA below:

**Motivation**

---

[*]Equal Contribution

Submitted to the 38th Conference on Neural Information Processing Systems (NeurIPS 2024) Track on Datasets and Benchmarks. Do not distribute.

Q1. **For what purpose was this dataset created?** *Was there a specific task in mind? Was there a specific gap that needed to be filled? Please provide a description.*

We aim to address the limitations of current Visual Question Answering (VQA) datasets, which predominantly focus on English and Western-centric images. These datasets lack diversity, especially in low-resource languages and culturally varied images. To overcome these issues, we introduce CVQA, a new benchmark designed to include culturally-driven images and questions from 28 countries covering 26 languages and 11 scripts. This benchmark aims to enhance the evaluation of multimodal AI models, encouraging the development of models with better cultural awareness and linguistic diversity.

Q2. **Who created the dataset (e.g., which team, research group) and on behalf of which entity (e.g., company, institution, organization)?**

CVQA is a collaborative movement involving many people from different institutions and communities. The CVQA is led by a team of researchers from MBZUAI.

Q3. **Who funded the creation of the dataset?** *If there is an associated grant, please provide the name of the grantor and the grant name and number.*

No grant, all expenses were funded by the MBZUAI's faculty startup fund.

Q4. **Any other comments?**

No.

**Composition**

Q5. **What do the instances that comprise the dataset represent (e.g., documents, photos, people, countries)?** *Are there multiple types of instances (e.g., movies, users, and ratings; people and interactions between them; nodes and edges)? Please provide a description.*

We provide a test set that contains instances of image-question pairs. Specifically, each instance is a dictionary that contains: `image, ID, Subset, Question, Translated Question, Options, Translated Options, Label, Category, Image Type, Image Source, License`. We provide a more detailed description of each field and an example in the README of `https://huggingface.co/datasets/afaji/cvqa`.

Q6. **How many instances are there in total (of each type, if appropriate)?**

CVQA contains a `test-split` of 9044 instances of image-question pairs.

Q7. **Does the dataset contain all possible instances or is it a sample (not necessarily random) of instances from a larger set?** *If the dataset is a sample, then what is the larger set? Is the sample representative of the larger set (e.g., geographic coverage)? If so, please describe how this representativeness was validated/verified. If it is not representative of the larger set, please describe why not (e.g., to cover a more diverse range of instances, because instances were withheld or unavailable).*

Yes, it contains all instances.

Q8. **What data does each instance consist of?** *"Raw" data (e.g., unprocessed text or images) or features? In either case, please provide a description.*

We provide raw annotations, where each instance consists of an image, a question and four answer candidates - of which only one is correct.

Q9. **Is there a label or target associated with each instance?** *If so, please provide a description.*

Yes, we provide four answer candidates for each question; among each set of four, we have labelled one as correct.

Q10. **Is any information missing from individual instances?** *If so, please provide a description, explaining why this information is missing (e.g., because it was unavailable). This does not include intentionally removed information, but might include, e.g., redacted text.*

No.

**Q11. Are relationships between individual instances made explicit (e.g., users' movie ratings, social network links)?** *If so, please describe how these relationships are made explicit.*

Some of the images are associated with more than one question (max three).

**Q12. Are there recommended data splits (e.g., training, development/validation, testing)?** *If so, please provide a description of these splits, explaining the rationale behind them.*

All data is for test purposes.

**Q13. Are there any errors, sources of noise, or redundancies in the dataset?** *If so, please provide a description.*

The data is human-written so it is bound to errors such as typos or grammatical errors. However, we argue that these errors are naturally made and (in very small amounts) are good for benchmarking the model's robustness, as the purpose of creating the dataset.

**Q14. Is the dataset self-contained, or does it link to or otherwise rely on external resources (e.g., websites, tweets, other datasets)?** *If it links to or relies on external resources, a) are there guarantees that they will exist, and remain constant, over time; b) are there official archival versions of the complete dataset (i.e., including the external resources as they existed at the time the dataset was created); c) are there any restrictions (e.g., licenses, fees) associated with any of the external resources that might apply to a future user? Please provide descriptions of all external resources and any restrictions associated with them, as well as links or other access points, as appropriate.*

It is self-contained in `https://huggingface.co/datasets/afaji/cvqa`

**Q15. Does the dataset contain data that might be considered confidential (e.g., data that is protected by legal privilege or by doctor–patient confidentiality, data that includes the content of individuals' non-public communications)?** *If so, please provide a description.*

No.

**Q16. Does the dataset contain data that, if viewed directly, might be offensive, insulting, threatening, or might otherwise cause anxiety?** *If so, please describe why.*

No.

**Q17. Does the dataset relate to people?** *If not, you may skip the remaining questions in this section.*

No.

**Q18. Does the dataset identify any subpopulations (e.g., by age, gender)?**

By country.

**Q19. Is it possible to identify individuals (i.e., one or more natural persons), either directly or indirectly (i.e., in combination with other data) from the dataset?** *If so, please describe how.*

No.

**Q20. Does the dataset contain data that might be considered sensitive in any way (e.g., data that reveals racial or ethnic origins, sexual orientations, religious beliefs, political opinions or union memberships, or locations; financial or health data; biometric or genetic data; forms of government identification, such as social security numbers; criminal history)?** *If so, please provide a description.*

Yes - ethnic origins. The purpose of this dataset is to capture culture in the images. However, as public faces are blurred/ unrecognisable, this feature does not pose any risk.

**Q21. Any other comments?**

No.

**Collection Process**

**Q22. How was the data associated with each instance acquired?** *Was the data directly observable (e.g., raw text, movie ratings), reported by subjects (e.g., survey responses), or indirectly inferred/derived from other data (e.g., part-of-speech tags, model-based guesses for age or language)? If data was reported by subjects or indirectly inferred/derived from other data, was the*

*data validated/verified? If so, please describe how.*

The image is obtained from either web or self-made images, we selected web images with Creative Commons license. Questions and options are written by annotators.

**Q23. What mechanisms or procedures were used to collect the data (e.g., hardware apparatus or sensor, manual human curation, software program, software API)?** *How were these mechanisms or procedures validated?*

Manual human curation. See Section 5 and 4 for details about our annotation platform and guideline. For more information, please refer to our paper.

**Q24. If the dataset is a sample from a larger set, what was the sampling strategy (e.g., deterministic, probabilistic with specific sampling probabilities)?**

N/A

**Q25. Who was involved in the data collection process (e.g., students, crowdworkers, contractors) and how were they compensated (e.g., how much were crowdworkers paid)?**

Those involved in the data collection process have been named as co-authors.

**Q26. Over what timeframe was the data collected? Does this timeframe match the creation timeframe of the data associated with the instances (e.g., recent crawl of old news articles)?** *If not, please describe the timeframe in which the data associated with the instances was created.*

The data was collected from 2023 to 2024.

**Q27. Were any ethical review processes conducted (e.g., by an institutional review board)?** *If so, please provide a description of these review processes, including the outcomes, as well as a link or other access point to any supporting documentation.*

Ethical Review was not required for this dataset collection.

**Q28. Does the dataset relate to people?** *If not, you may skip the remaining questions in this section.*

No, Annotators were asked to provide data about their country's culture but not specific to any individual or groups of people.

**Q29. Did you collect the data from the individuals in question directly, or obtain it via third parties or other sources (e.g., websites)?**

N/A.

**Q30. Were the individuals in question notified about the data collection?** *If so, please describe (or show with screenshots or other information) how notice was provided, and provide a link or other access point to, or otherwise reproduce, the exact language of the notification itself.*

N/A.

**Q31. Did the individuals in question consent to the collection and use of their data?** *If so, please describe (or show with screenshots or other information) how consent was requested and provided, and provide a link or other access point to, or otherwise reproduce, the exact language to which the individuals consented.*

N/A.

**Q32. If consent was obtained, were the consenting individuals provided with a mechanism to revoke their consent in the future or for certain uses?** *If so, please provide a description, as well as a link or other access point to the mechanism (if appropriate).*

N/A.

**Q33. Has an analysis of the potential impact of the dataset and its use on data subjects (e.g., a data protection impact analysis) been conducted?** *If so, please provide a description of this analysis, including the outcomes, as well as a link or other access point to any supporting documentation.*

N/A.

**Q34. Any other comments?**

No.

**Preprocessing, Cleaning and/or Labeling**

Q35. **Was any preprocessing/cleaning/labeling of the data done (e.g., discretization or bucketing, tokenization, part-of-speech tagging, SIFT feature extraction, removal of instances, processing of missing values)?** *If so, please provide a description. If not, you may skip the remainder of the questions in this section.*

Yes, we removed all image metadata, automatically blurred faces and text that would reveal the answer, and finally removed images that had invalid licenses.

Q36. **Was the "raw" data saved in addition to the preprocessed/cleaned/labeled data (e.g., to support unanticipated future uses)?** *If so, please provide a link or other access point to the "raw" data.*

No.

Q37. **Is the software used to preprocess/clean/label the instances available?** *If so, please provide a link or other access point.*

We used a standard Python code to clean and preprocess the final instances of CVQA. During the data collection process we allowed the annotators to use "PicdeFacer", a tool that can be used for blurring faces or information: `https://picdefacer.com/en/`.

Q38. **Any other comments?**

No.

**Uses**

Q39. **Has the dataset been used for any tasks already?** *If so, please provide a description*

Our dataset have not been used for other tasks yet. We only use it in our paper to benchmark various models.

Q40. **Is there a repository that links to any or all papers or systems that use the dataset?** *If so, please provide a link or other access point.*

No.

Q41. **What (other) tasks could the dataset be used for?**

Primarily for benchmarking Cultural multilingual visual QA, but this dataset can potentially be used for Machine Translation and language learning game (CALL).

Q42. **Is there anything about the composition of the dataset or the way it was collected and preprocessed/cleaned/labeled that might impact future uses?** *For example, is there anything that a future user might need to know to avoid uses that could result in unfair treatment of individuals or groups (e.g., stereotyping, quality of service issues) or other undesirable harms (e.g., financial harms, legal risks) If so, please provide a description. Is there anything a future user could do to mitigate these undesirable harms?*

No.

Q43. **Are there tasks for which the dataset should not be used?** *If so, please provide a description*

No.

Q44. **Any other comments?**

No.

**Distribution**

Q45. **Will the dataset be distributed to third parties outside of the entity (e.g., company, institution, organization) on behalf of which the dataset was created?** *If so, please provide a description.*

Yes, the data has been publicly released.

206 Q46. **How will the dataset be distributed (e.g., tarball on website, API, GitHub)?** *Does the*
207 *dataset have a digital object identifier (DOI)?*
208 The data is available on Huggingface at: `https://huggingface.co/datasets/afaji/cvqa`.

209 Q47. **When will the dataset be distributed?**
210 CVQA is already available from June 2024 and onward.

211 Q48. **Will the dataset be distributed under a copyright or other intellectual property (IP) license,**
212 **and/or under applicable terms of use (ToU)?** *If so, please describe this license and/or ToU, and*
213 *provide a link or other access point to, or otherwise reproduce, any relevant licensing terms or ToU,*
214 *as well as any fees associated with these restrictions.*
215 Note that each instance has its own license. All data is free to use for research purposes, but not
216 every entry is permissible for commercial use.

217 Q49. **Have any third parties imposed IP-based or other restrictions on the data associated with**
218 **the instances?** *If so, please describe these restrictions, and provide a link or other access point*
219 *to, or otherwise reproduce, any relevant licensing terms, as well as any fees associated with these*
220 *restrictions.*
221 Yes, some images on Flickr, are under copyright. We advised annotators to only select those
222 available for non-commercial use: `https://creativecommons.org/licenses/by-nc-nd/4.0/`
223 `deed.en`. We also automatically remove entries that do not conform to the copyright requirement.

224 Q50. **Do any export controls or other regulatory restrictions apply to the dataset or to individual**
225 **instances?** *If so, please describe these restrictions, and provide a link or other access point to, or*
226 *otherwise reproduce, any supporting documentation.*
227 No.

228 Q51. **Any other comments?**
229 No.

230 **Maintenance**
231
232 Q52. **Who will be supporting/hosting/maintaining the dataset?**
233 CVQA team at MBZUAI.

234 Q53. **How can the owner/curator/manager of the dataset be contacted (e.g., email address)?**
235 You can contact the main team via email or through starting a new discussion on the CVQA Hugging
236 Face page.

237 Q54. **Is there an erratum?** *If so, please provide a link or other access point*
238 N/A.

239 Q55. **Will the dataset be updated (e.g., to correct labeling errors, add new instances, delete**
240 **instances)?** *If so, please describe how often, by whom, and how updates will be communicated to*
241 *users (e.g., mailing list, GitHub)?*
242 Yes, updates will be made on Huggingface once we have more data (e.g. new country-language
243 pairs) or there are reported errors in the data.

244 Q56. **If the dataset relates to people, are there applicable limits on the retention of the data**
245 **associated with the instances (e.g., were individuals in question told that their data would be**
246 **retained for a fixed period of time and then deleted)?** *If so, please describe these limits and explain*
247 *how they will be enforced.*
248 N/A.

249 Q57. **Will older versions of the dataset continue to be supported/hosted/maintained?** *If so,*
250 *please describe how. If not, please describe how its obsolescence will be communicated to users.*
251 N/A.

252 Q58. **If others want to extend/augment/build on/contribute to the dataset, is there a mechanism**
253 **for them to do so?** *If so, please provide a description. Will these contributions be validated/verified?*

254 *If so, please describe how. If not, why not? Is there a process for communicating/distributing these*
255 *contributions to other users? If so, please provide a description.*
256 Yes, it will be conducted through communications with CVQA team.

257 Q59. **Any other comments?**
258 No.

## 2 Data Access and Maintenance Plan

260 We publicly released CVQA, and it is available to download from Hugging Face: `https:`
261 `//huggingface.co/datasets/afaji/cvqa`. To assess model performance, we also created a
262 leaderboard in `eval.ai` platform: `https://eval.ai/web/challenges/challenge-page/2305/`
263 `overview`. Detailed information is provided in `www.cvqa-benchmark.org`. We, the authors, will be
264 responsible for handling CVQA issues and maintaining the data accordingly.

## 3 Statement of Responsibility

266 We, the authors, bear all responsibilities in case of rights violations in CVQA. Please note that each
267 image question in our dataset has its own distinct license. Although our data is free to use for research
268 purposes, not all instances in CVQA are permissible for commercial use.

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

 2) and select the entry to see a detailed preview of the submission (Figure 3), here they can edit the data directly, provide comments, or confirm the data by starring the entry.

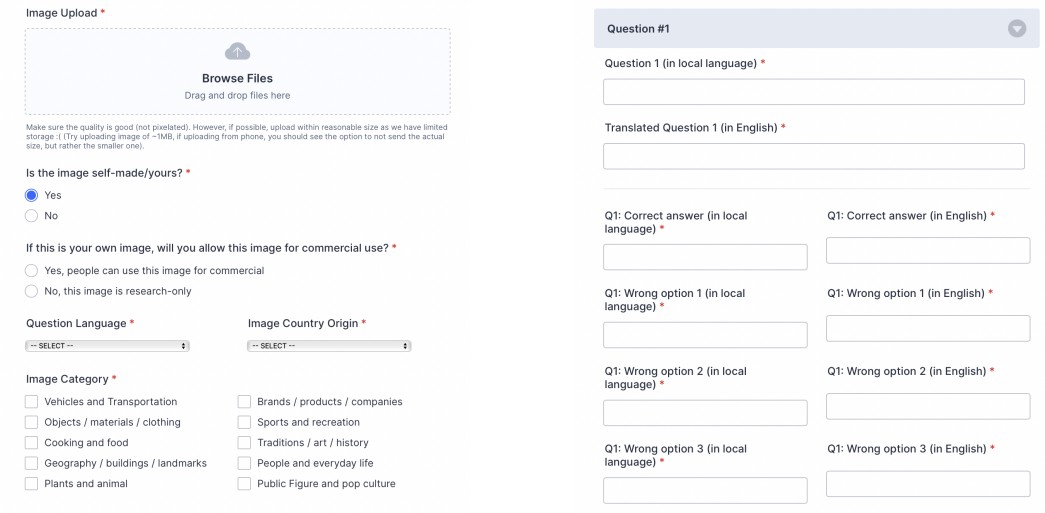

Figure 1: Annotation interface for entering image and questions

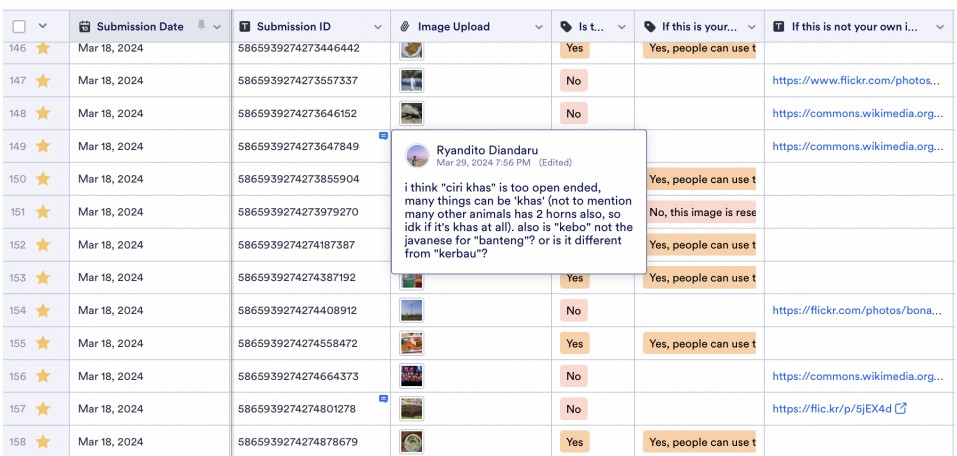

Figure 2: Annotation interface for validation. Contributors can comment, edit, and star the entries

## 6 Additional Statistics

### 6.1 Most-Frequent Words in the Questions

Figure 4 shows word clouds for the most frequent words in CVQA per category. We exclude stopwords as well as 'picture', 'photo', and 'image' from the list, since most questions contain these words. In this VQA context, we can treat them as stopwords.

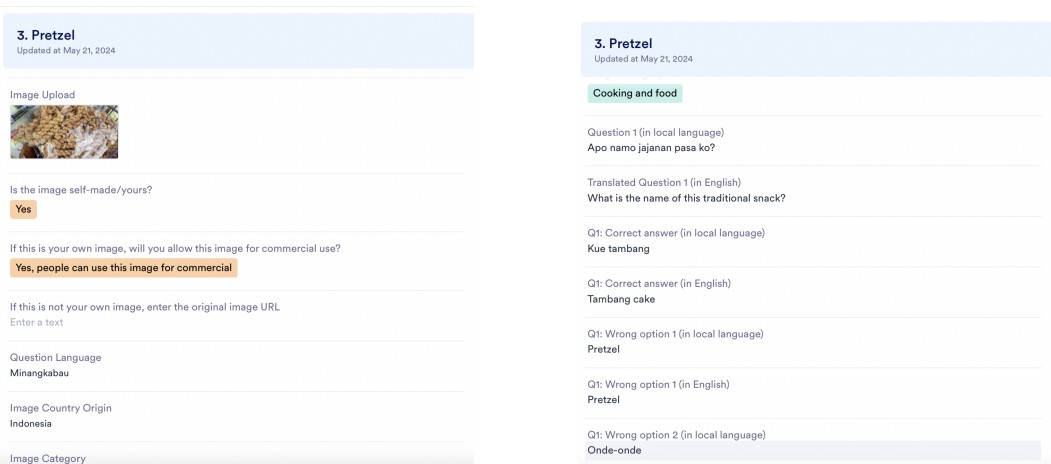

Figure 3: During validation, contributors can preview the submission from other contributors

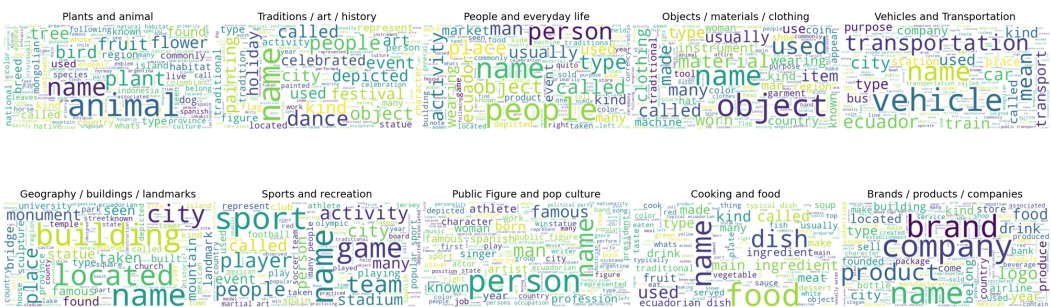

Figure 4: Word Cloud in CVQA per category

## 6.2 CVQA Annotator Demographic

Figure 5 illustrates the demographic statistics of the annotators, based on an anonymous questionnaire we provided. At the time of writing, we have information for 36 out of 76 annotators. As such, this breakdown is a rough representation of the annotation group.

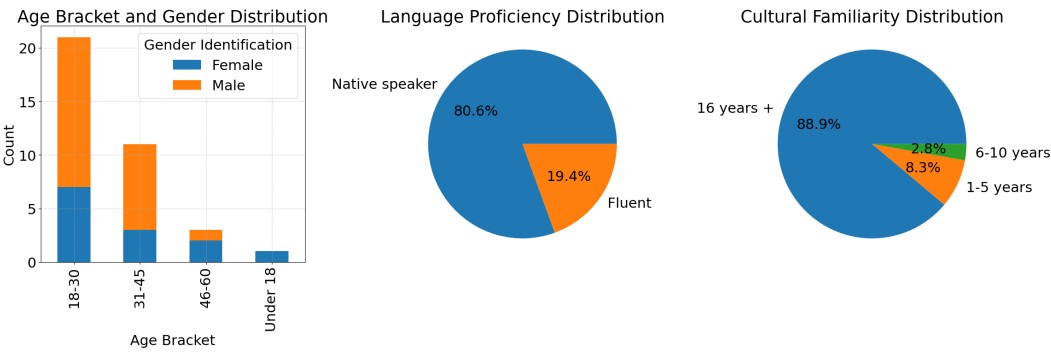

Figure 5: Annotator demographic statistics

## 6.3 Country-Language Pairs and Scripts

In Table 1, we provide information on the script used in each Country-Language pair.

| Country | Language | Script |
|---------|----------|--------|
| *Africa* | | |
| Egypt | Egyptian Arabic | Arabic |
| Ethiopia | Amharic | Amharic |
| Ethiopia | Oromo | Latin |
| Nigeria | Igbo | Latin |
| *Asia* | | |
| China | Chinese | Chinese |
| India | Bengali | Bengali |
| India | Tamil | Tamil |
| Indonesia | Indonesian | Latin |
| Indonesia | Javanese | Latin |
| Indonesia | Minangkabau | Latin |
| Indonesia | Sundanese | Latin |
| Japan | Japanese | Japanese |
| South Korea | Korean | Hangul |
| Malaysia | Malay | Latin |
| Mongolia | Mongolian | Cyrillic |
| Pakistan | Urdu | Perso-Arabic |
| Philippines | Filipino | Latin |
| Singapore | Chinese | Chinese |
| Sri Lanka | Sinhala | Sinhalese |
| *Europe* | | |
| Bulgaria | Bulgarian | Cyrillic |
| France | Breton | Latin |
| Ireland | Irish | Latin |
| Norway | Norwegian | Latin |
| Romania | Romanian | Latin |
| Russia | Russian | Cyrillic |
| Spain | Spanish | Latin |
| *Latin America* | | |
| Argentina | Spanish | Latin |
| Brazil | Portuguese | Latin |
| Chile | Spanish | Latin |
| Colombia | Spanish | Latin |
| Ecuador | Spanish | Latin |
| Mexico | Spanish | Latin |
| Uruguay | Spanish | Latin |

Table 1: The list of Country-Language pairs covered in CVQA and their corresponding scripts.

# References

[1] T. Gebru, J. Morgenstern, B. Vecchione, J. W. Vaughan, H. Wallach, H. D. I. au2, and K. Crawford. Datasheets for datasets, 2021.