# OpenReview forum: "CVQA: Culturally-diverse Multilingual Visual Question Answering Benchmark"
_NeurIPS.cc/2024/Datasets_and_Benchmarks_Track — NeurIPS 2024 Track Datasets and Benchmarks Oral_

### Official Review · Reviewer_mHUB · 2024-07-03
**A multiple-choice VQA benchmark spanning 28 countries, 4 continents, 26 languages and 11 scripts**

**Rating:** 7
**Confidence:** 4

**Review:**

Overall, this paper is commendable and addresses the critical absence of VQA datasets and benchmarks representing non-Western culture and non-English languages. The dataset was collected using a sound and systematic methodology, which has been well-documented, and spans a wide suite of cultures/languages compared to previous works. While the dataset itself is valuable, the benchmark is primarily formulated as a 4-way multiple choice task which is simplistic and may not necessarily drive progress towards truly culturally-inclusive MLLMs. The authors address this by saying that the dataset does lend itself to a more open-ended generation task, however the authors only briefly explore this in the paper. The accompanying baselines, while not novel in themselves, are thorough and provide interesting insights into the capabilities of current models. They also provide a necessary starting point for future research in this direction.

Please see Strengths and Opportunities sections below for further comments.

**Strengths:**

- The dataset spans a large suite of regions, languages and scripts -- larger than prior works have considered.
- The dataset is collected using a sound and systematic methodology, with annotators having lived experience in the countries/languages that are considered. Annotators from those regions are also then used to validate the samples that are collected.
- The paper situates itself well within the broader ongoing discussion in the literature around what it means to "measure culture" (i.e. following Adilazuarda et al. and using common-ground knowledge as a proxy of culture)
- The benchmark and associated metrics, which focus on a 4-way multiple-choice task, are well constructed and described.
- The baselines span a relevant selection of current models, including GPT4o which was released only very recently.
- The paper is well written and structured

**Additional Feedback:**

Are all questions in the dataset guaranteed to be culturally related? One of the examples given in the paper ("What is the color of the t-shirt the youngest member of this group is wearing?") does not appear to be so.

**Clarity:**

The paper is well written, well structured, and easy to read. All relevant information to understand the dataset's collection and benchmark results were included, with appropriate pointers to the appendices where needed.

Some places which could be better worded/described:
- L196 "Additionally, due to the multilingual nature of CVQA, for each prompt, we evaluate using the English-only and local language question-option pairs". This was difficult to understand without looking at the tables.
- L260 "Then, the answer is selected by choosing the model’s highest probability of generating the full answer phrase of one of the options [11] (e.g., Jakarta, Bandung, Bali, Surabaya)." This could be described in more detail to facilitate better understanding.

**Correctness:**

The contributions claimed in the paper are sound and supported with appropriate empirical evidence.

**Documentation:**

The dataset is sufficiently documented. There is a leaderboard and project page, where the data can be accessed. It is hosted on Hugging Face.

**Ethics:**

Given the somewhat limited quality control used when validating the data, there is a chance that PII may be present in this dataset.
Other than this, I do not suspect any significant ethical concerns.

**Limitations:**

The authors provide a good discussion of their paper's limitations.
One aspect which is not addressed (as far as I could see) was what license will be attached to the dataset and how annotators are compensated for their data (beyond paper authorship for super-annotators)

**Opportunities For Improvement:**

- The benchmark's primary evaluation setting is a 4-way multiple-choice task. This is simplistic and not reflective of most real-world scenarios in which VQA would be used. While it provides a 'starting point' for studying culturally-diverse VQA, having the research community "adopt" and optimise performance on this benchmark may not guarantee that we ultimately achieve culturally-inclusive multi-modal models.
- The authors duly note the above and highlight that the dataset can also be used in an open-ended generation evaluation, which they explore briefly in the paper. However, this section is quite limited (with no pointers to results tables). Their performance metric under this setting (computing the probability of each of the multiple-choice answers under the model) also does not allow for paraphrased answers which is another highly realistic scenario which may arise. Works like Manas et al (AAAI 2024) have taken steps towards measuring semantic similarity between a generated answer and a set of reference answers.
- The authors note that "the multiple-choice setting is often brittle towards option ordering" but provide no ablation/analyses which investigates if this is the case within their experiments. This should ideally be included.
- While the authors note that the annotators were "trained", the extent of their training seems to be a set of written guidelines. It is not necessarily guaranteed that annotators will always understand or follow these. While the authors do include a validation phase, each sample is only checked by 1 other annotator which may not catch sub-par examples. Was any further QC/checks done on the dataset?
 - The inclusion of CLIP and mCLIP among the baseline models does not feel well justified, given that these models are not VQA models and could not be extended to an open-generation setting. While it may not make sense to remove these experiments since they've been run, it might be useful to include a sentence or two on why they were selected.
- Closed source models (Gemini and GPT4o) are only included for part of the analyses. It would be good to have all the analyses extended to these models, if costs permit.

**Relation To Prior Work:**

The paper provides sufficient evidence to distinguish its contributions from prior works -- specifically, CVQA includes images _and_ questions that have been collected by annotators who have lived experience in a wide set of different cultures. All prior works have either 1) used Western-centric images and translated their associated visual questions into different languages hence do not necessarily capture culturally-specific scenes, or 2) focused on uni-modal settings (i.e. pure text questions, without images, collected from different cultures). CVQA also spans a wider set of cultures (26) compared to other works.

**Summary And Contributions:**

The paper introduces a new VQA benchmark - Culturally-diverse VQA (CVQA) - which includes ~9k visual questions collected across 28 countries, 4 continents, 26 languages and 11 scripts. The images and questions were collected (and validated) from communities within these regions, and include both local language and English translations for each question. The paper then benchmarks 8 multi-modal language models (MLLMs) on their benchmark, showcasing that state-of-the-art VQA models exhibit poor performance on culturally-nuanced visual questions.

Their contributions are:
- A dataset of ~9k visual questions spanning 28 countries and 26 languages, including documentation of the dataset's contents and collection
- A suite of baselines on the benchmark, using current state-of-the-art MLLMs

---

> ### Author Rebuttal · Authors · 2024-08-15
>
> Thank you for your detailed review and recognition of our dataset's value in addressing the lack of culturally diverse VQA benchmarks. We appreciate your insights and will address your comments as follows:
>
> **1. Multiple-choice task limitations:**
> We acknowledge your concern about the simplicity of the 4-way multiple-choice task. While it serves as an accessible starting point, we agree that it may not fully drive progress towards truly culturally-inclusive MLLMs. During the creation of our CVQA dataset, we made sure that CVQA is also convertible into free-text open-ended QA - see line 125. We also conducted experiments examining the performance of MLLMs without option list in the input (free-generation QA), see line 255-266.  We'll also discuss future work directions that could involve more complex evaluation metrics, such as those proposed by Manas et al. (AAAI 2024) for measuring semantic similarity in generated answers.
>
> **2. Option ordering robustness:**
> We appreciate this observation. We have already shuffled the options in the original dataset to make sure that the option order is random. We will conduct an ablation study on option ordering and include the results in the paper, providing insights into the robustness of our benchmark to this factor.
>
> **3. Annotator training and quality control:**
> We will provide more details on the annotator training process, including any additional quality control measures beyond the initial validation. We have conducted further quality checks by several iterations to ensure the quality of the annotations and that they follow the initial guidelines of this project, which are performed by the core contributors of this project.
>
> **4. Justification for CLIP and mCLIP:**
> We included CLIP and mCLIP as simple yet strong embedding-based baselines to provide a comparison point for more complex VQA models. While they are not full VQA models, their performance on this task offers valuable insights into the cross-modal understanding capabilities of widely-used vision-language models. In the revised paper, we will clarify their role as baselines and discuss how their performance compares to more specialized VQA models, highlighting the strengths and limitations of embedding-based approaches in culturally diverse contexts.
>
> **5. Closed-source model analyses:**
> We will try to extend all analyses to include Gemini and GPT4o and explain any limitations of these models.
>
> **6. Dataset license and annotator compensation:**
> We can confirm that all images from public sources are under Creative Commons license and are at least permitted for research use. We will add more detailed information about the dataset's license. As this is a community-driven research project, we offered authorship as compensation for contributors rather than monetary compensation. This provides valuable academic credit. All contributors agreed to this arrangement and recognise the value of authorship
>
> Thank you for your thorough review and valuable suggestions for improving our paper's comprehensiveness and impact. We hope that our responses have addressed your concerns and that you will consider increasing your score.

---

> > ### Comment · Reviewer_mHUB · 2024-08-16
> >
> > Thanks for the responses, authors.
> >
> > Regarding point 1, you mentioned that you have provided results on the open-ended generation task in L255-L266 however these are just written in text. Is there a table somewhere which compares the open-ended generation results across all the models (and disaggregated by the different image types) - similar to Tables 5-7?
> >
> > All my other comments have been addressed satisfactorily.

---

> > > ### Author Rebuttal · Authors · 2024-08-21
> > >
> > > Thank you for pointing out the open-ended generation results. Below are the open-ended generation evaluation resutls of LLaVA-1.5-7B disaggregated by image type and question type respectively:
> > >
> > > | Prompt Type | Self-made Image | Web Image |
> > > |----------------|-------------------|------------------|
> > > | English Prompts   | 35.4      | 37.3      |
> > > | Multilingual Prompts   | 28.3      | 32.3     |
> > >
> > >
> > > | **Prompt Type**         | **Vehicles** | **Geography** | **Tradition** | **Plants & Animal** | **Food** | **Pop Culture** | **Objects** | **Sports** | **People** | **Brands** |
> > > |-------------------------|--------------------|---------------|---------|---------------------|----------|-----------------|-------------|------------|-------------------|------------|
> > > | **English Prompts**     | 38.1               | 37.9          | 33.6    | 39.5                | 32.0     | 42.0            | 32.5        | 36.2       | 39.1              | 40.6       |
> > > | **Multilingual Prompts**| 30.8               | 32.4          | 32.2    | 30.6                | 27.2     | 34.6            | 27.8        | 31.9       | 27.4              | 31.1       |
> > >
> > >
> > > We will include these results in the revised version with extra pages.

---

> > > > ### Comment · Reviewer_mHUB · 2024-08-21
> > > >
> > > > Thanks. This has addressed all my concerns.

---

### Official Review · Reviewer_Hjkn · 2024-07-22
**New Dataset for VQA Includes Culturally Diverse Questions and Images**

**Rating:** 7
**Confidence:** 5

**Review:**

The paper is easy to read and generally includes a lot of detail regarding the dataset and how it was collected. The contribution is significant with the

**Strengths:**

The dataset's premise is really good to see in a world in which AI models are becoming commonplace - so to ensure that they are applicable to a larger variety of countries, cultures, and languages. The experiments include a good amount of discussion and showcase interesting aspects of the new benchmark.

There are also good ablation studies/auxiliary experiments comparing the performance on each of the different categories, the source of the images (comparing web to self-made images), and also to location-aware images.

**Additional Feedback:**

N/A

**Clarity:**

The paper is well written and includes a lot of information throughout - it is an easy read.

**Correctness:**

Yes, the dataset looks to be constructed in a sound way with a lot of attention to detail within the community as a useful evaluation benchmark. Experiments are performed correctly.

**Documentation:**

The paper includes a datasheets for datasets sheet which includes all necessary detail including a url link to the dataset itself (present also on the first page of the paper).

**Ethics:**

Concerns regarding:
* Consent of images which were gathered from common.wikimedia.org, Flickr, GapMinder, Unsplash, Pixabay - were the original authors of the images asked for their permission?
* Fair wages, it is mentioned that contributors were rewarded with authorship on the paper, but there is no mention of monetary reimbursement for their time or effort. What is the effort expended here?
* Research involving human participants: participants were often using their own images and it is unclear whether this has passed through an ethics review board, IRB, or similar.

**Limitations:**

Yes, the authors have addressed the limitations within the work, mentioning that whilst this is meant to be a diverse collection, there is no way that it can cover everything, it is "only" 28 countries and 26 languages which is a far cry from the number of countries/languages in the world.

**Opportunities For Improvement:**

One experiment which may be interesting to see is how well the models perform on this dataset if the original questions are translated to English and then asked into the models. For example, new columns in Table 3 which includes the answer from an LLM where the prompt has been auto-translated to English.

The paper generally lacks qualitative examples and/or example images and questions/answer sets from the dataset itself. Figure 1 includes 4 examples from the dataset, but otherwise the only other examples are in the annotation information within the appendix and it's not clear if these are images from the actual dataset or not. Given the uniqueness and potential usefulness of the dataset it would be good to see some more examples.

Another experiment which is missing is a text-only baseline. Within the collection information, it asks collectors to ensure that questions should require the image to answer the question, but it would be good to see results on this to verify that this aspect of the paper has been captured or not.

**Relation To Prior Work:**

The related work section is thorough and includes both normal VQA datasets and how they have changed as well as multi-lingual VQA datasets. It highlights a potential language bias in using English for everything to help motivate the paper.

**Summary And Contributions:**

This paper proposes a new dataset, Culturally-diverse, multi-lingual Visual Question Answering Benchmark (CVQA). CVQA includes a large number of images as well as questions in different languages, cultures, and countries, including 28 countries and 26 languages.
The dataset includes all questions in both English and the collector's other language. Experiments showcase the difficulty of the dataset for even state of the art Vision Language Models in languages other than English with large drops in performance comparing questions in English with the other language.

---

> ### Author Rebuttal · Authors · 2024-08-15
>
> Thank you for your positive feedback and thoughtful suggestions. We appreciate your recognition of our dataset's significance in promoting AI applicability across diverse cultures and languages.
>
> **1. Translated English experiment:**
> Your suggestion to include results from auto-translated English prompts is valuable. We will conduct this additional experiment and add the results to Table 3 to give more insights into the impact of translation on model performance.
>
> **2. More qualitative examples:**
> We agree that including more examples would enhance the paper. We will add a new figure with diverse examples from the dataset, showcasing various languages, cultures, and question types. This will give readers a better sense of the dataset's content and diversity.
>
> **3. Text-only baseline:**
> Thank you for this suggestion. Actually we conducted experiments of not providing image inputs (a black image instead) for multimodal LLMs (e.g. LLaVA), where we observe a substantial performance drop. We will also consider conducting some text-only baselines (like text-only LLMs) experiment to verify that the questions indeed require visual information. We'll include these results in a new subsection, demonstrating the visual dependency of our dataset.
>
> **4. Ethics concerns:**
> - Image consent: We confirm that all images from public sources were used in compliance with their respective licenses, all of them are under Creative Commons which permit such usage for research purposes. Later we also have a post-processing stage to crawl the image license to confirm that indeed all our images are permissible.
> - Fair wages: Our work follows prior community-based research such as BigScience (BLOOM, P3 instruction dataset), NusaCrowd, SEACrowd, Universal NER, and other dataset construction work where the annotators/contributors are involved with the research and rewarded with paper authorship. Annotators/authors are aware of this agreement before joining our project.
> - Research involving human participants: We will include details about the ethical review process undertaken for this study. Note that all contributors are involved with the research and co-authors in this paper, therefore, no IRB approval is needed.
>
> We appreciate your thorough review and suggestions for improving both the content and ethical considerations of our paper. We hope that our responses have addressed your concerns and that you will consider increasing your score.

---

> > ### Comment · Reviewer_Hjkn · 2024-08-16
> >
> > Thanks for responding to my questions it has cleared up many of my concerns. Regarding the ethics concerns, it is mentioned on line 80:
> > *"To promote data collection, contributors with significant contributions, either by contributing at least 100 validated question-answer pairs and/or managing several subsets, are rewarded as co-authors in this paper."*
> >
> > Does this mean that some contributors weren't rewarded with authorship if they provided less question answer pairs?

---

> > > ### Author Rebuttal · Authors · 2024-08-16
> > >
> > > - Correct. However, because of this (we tell this in advance), and that we motivate the onboarded data contributors to meet the required contribution, all contributors provided more than the threshold. In other words, there are no covered languages in CVQA with data contributors who provided ‘just under the threshold’ (e.g., 95 questions).
> > >
> > > - There are few languages that only contributes ~5 questions or so before being inactive/dropping the project, so in that case we had to drop the whole language. In other words, either contributors only attempted to add ~5 questions before dropping out from the project, or they will be fully contributing to the project and provided 100+ questions.

---

> > > > ### Comment · Reviewer_Hjkn · 2024-08-16
> > > >
> > > > Thanks for the clarification. It might be worth updating the text within the paper to make this distinction clear, as from reading it, it can be assumed that there may be many contributors who only provided 5-95 images and therefore did not receive any compensation for their time/effort.

---

> > > > > ### Author Rebuttal · Authors · 2024-08-18
> > > > >
> > > > > Thanks for the suggestion. We will include anonymous statistics of onboarded contributors who dropped out of the project.

---

### Official Review · Reviewer_9Fd4 · 2024-07-22
**Great VQA datasets for various languages and cultures.**

**Rating:** 9
**Confidence:** 4
**Correctness:** Yes
**Clarity:** Yes

**Review:**

Please see the summary, strengths, and Opportunities For Improvement.

**Strengths:**

1. A dataset that is not limited to Western images. Constructed data reflecting various languages ​​and cultures by native speakers and cultural experts of various languages/cultures (28 countries, 26 languages, 10 categories)

2. There are hundreds of samples for each language, which is also suitable for a cultural VQA evaluation for a single language.

3. Great documentation (including supplementary materials) on data collection and annotation. They can be helpful when creating similar data in the future (for example, when trying to increase languages ​​or categories).

4. Each data has a question/answer in English (EN) and local languages ​​(LOC), and experiments conducted with EN and LOC prompts provide interesting insights.

**Additional Feedback:**

None

**Documentation:**

Yes

**Limitations:**

Yes

**Opportunities For Improvement:**

1. With CVQA, conducting quantitative evaluation seems easy, but qualitative analysis seems difficult because there is no answer information for each question (in Huggingface's data, all answers were -1).
Will the answer information be released later? Or are the authors planning to not release it?
It's easy to figure out the answers to questions about familiar languages and cultures, but it's hard to know the answers to questions about unfamiliar languages/cultures.
So, I thought that CVQA would be a difficult dataset to do qualitative analysis if the answers were not made public.
Like MMMU's dev or val set, it would be easy to use if at least some of the answers were made public.

2. As far as I know, Hindi is the representative language of India, but Table 8 (=Table 1 in supplementary materials) does not contain the Hindi language.
Is it okay to not have Hindi? Is there a reason why CVQA does not contain Hindi?
Likewise, it is curious that there is no French for France in Table 8.

3. Enlarging the radar part in Figure 3 a little more would be good. There is a lot of empty space now, and the graph is small, so it is a bit difficult to see.
Specifically, because there are so many languages, it is a bit difficult to see which point corresponds to which language.

4. From the 16p, supplementary materials suddenly appear...

**Relation To Prior Work:**

Yes

**Summary And Contributions:**

Most current VQA datasets focus on English or a few languages ​​for questions and focus on Western images.
Because of this, it is difficult to evaluate VQA capabilities for various languages ​​and cultures in the world.
Although translating the question to other languages can partially overcome this issue, the image often remains the same, which limits it to "VQA on Western culture."
To overcome this issue, the authors created a novel Culturally-diverse multi-lingual Visual Question Answering dataset (CVQA).
The authors created a new image (not only Western images) and hired native speakers and cultural experts to create CVQA that reflects various languages ​​and cultures.
CVQA is a challenging, high-quality new benchmark dataset.

---

> ### Author Rebuttal · Authors · 2024-08-15
>
> Thank you for your positive review and recognition of our dataset's value for various languages and cultures. We appreciate your detailed feedback and will address your concerns as follows:
>
> **1. Answer information for qualitative analysis:**
> We apologize for the confusion. For now, we do not plan to release the answers. We have launched a benchmark evaluation on: https://eval.ai/web/challenges/challenge-page/2305/overview, where practitioners can submit predictions and we will compare with the ground-truth answer to give the accuracy.
>
>
> **2. Hindi and French language inclusion:**
> Thank you for noting this important point. We acknowledge that Hindi and French are significant languages that would add value to our dataset. In our initial data collection, we aimed to maximize diversity across regions and language families while maintaining a minimum quality threshold. For this version, we set a minimum of 200 questions per country-language pair to ensure sufficient data for meaningful analysis.
> While we did collect some Hindi data, it did not meet this threshold in the current iteration. Similarly, for France, we prioritized regional languages to enhance linguistic diversity. However, we recognize the importance of including major languages like Hindi and French. We are actively working on expanding our dataset to include these languages in future versions. In the revised paper, we will clarify our language selection criteria and discuss our plans for future expansions to address these gaps.
>
>
> **3. Enlarging radar chart in Figure 3:**
> We agree that the current size makes it difficult to discern details. We will enlarge the radar chart portion of Figure 3 and adjust the layout to improve readability, ensuring that language labels are clearly visible.
>
> **4. Supplementary materials formatting:**
> We apologize for the abrupt transition to supplementary materials. We will add a clear section break and introductory text to improve the flow between the main paper and supplementary content.
>
> Thank you for your careful review and suggestions to enhance the paper's clarity and comprehensiveness. We hope that our responses have addressed your concerns and that you will consider increasing your score.

---

> > ### Comment · Reviewer_9Fd4 · 2024-08-27
> >
> > Thank you for the response.
> > I am still concerned that the researcher would have difficulty using this dataset since the answer information will not be released.
> > It would be helpful if at least part of the answer were released, but it is not a must.

---

> > > ### Author Response · Authors · 2024-08-28
> > > **We will release a validation set with labels**
> > >
> > > Thanks for your comments. We do have a plan to annotate a validation set with labels to facilitate and support the research on our CVQA dataset, we will release it on our benchmark website and Huggingface once it's finished.

---

> > > > ### Comment · Reviewer_9Fd4 · 2024-08-29
> > > >
> > > > Thank you for the response. It solved my biggest concern about the dataset, and thus, I raised my rating. Thank you for the great work!

---

### Official Review · Reviewer_nsfT · 2024-07-26
**Great resource and well-written paper**

**Rating:** 9
**Confidence:** 4
**Correctness:** Yes.
**Clarity:** Very clear.

**Review:**

Overall the paper was a great read, and introduces a very useful benchmark dataset. While there are a lot of new benchmarks being developed in the area of cultural evaluation of generative models, this dataset substantially adds to that line of work and resources. The work is also presented with generally good clarity (see areas of improvement below). While the approach of data collection is not novel, the significance of collected data, and the attention to meticulous details to ensure diversity, and the acknowledgement of limitations makes this still a great paper.

**Strengths:**

As outlined above, I believe that this will be an impactful resource for cultural evaluation of VQA models. The data may also help in evaluation of other multimodal models.

I love the care and attention to detail in the study design (e.g., design choices such as allowing to use own images vs. open-use images) as well as interpretation of results.

I especially appreciated the community oriented data collection approach (as opposed to collecting data through data annotation platforms, which introduces undesirable systemic biases). While this approach also introduces biases, the approach is more extensible. I also appreciate that data providers who contributed substantially to the dataset are also acknowledged as co-authors of this paper.

**Additional Feedback:**

None

**Documentation:**

Yes the data documentation seems pretty thorough, although the paper main text could benefit from more details.

**Ethics:**

No major concerns as such.

It does involve photos submitted by participants; but they were instructed to not use any faces etc., So the authors have taken reasonable precautions. But this is a sensitive area and I may have missed aspects that are more problematic.

**Limitations:**

Yes, the authors have provided a pretty meaningful discussion of limitations, given the context of the paper. Of course "culture" is a very complex topic and the discussion could be expanded further, but I think what they have included is sufficient. As suggested above, it may benefit from being brought up earlier.

**Opportunities For Improvement:**

It may be good to describe the complex nature of culture, and how difficult it is to capture in such static datasets (and similar other limitations you discuss in the limitations section) earlier on in the introduction itself so that the typical reader of this paper will also use the dataset with that larger context in mind. Currently it feels the complexities associated with the topic area is relegated to the Limitations section, which may be skipped by many readers. You may also want to point to literature from FAccT, AIES, CHI communities which has dealt with these questions in the computing context, or just broader literature in STS as background reading.

I think the annotation process could be better described in the main text. While the appendix has all the details, it still took a while for me to understand where the images came from and what were the instructions given to the annotators. Maybe a small paragraph or a brief bulleted list summarizing the annotation task might make it better. The current text in Section 2.2 assumes a level of understanding of VQA data collection approach that not all readers may be familiar with.

**Relation To Prior Work:**

Yes, the paper is largely well-situated in prior work.

**Summary And Contributions:**

The paper introduces a new Culturally-diverse multilingual Visual Question Answering benchmark that contains over 9000 questions collected from across 28 countries and 26 languages. The paper also demonstrate that the visual question answering performance of most publicly available models leave a lot of room for improvement, and that open models in particular have the most room for growth. Furthermore, their experiments also observe that model performance in native language is lower than in English, further pointing to another avenue for improvement.

---

> ### Author Rebuttal · Authors · 2024-08-15
>
> Thank you for your positive feedback and insightful comments. We appreciate your recognition of our dataset's potential impact and the care taken in its design.
>
> **1. Addressing cultural complexity earlier:**
> We agree with your suggestion to discuss the complex nature of culture and the challenges of capturing it in static datasets earlier in the paper. We will move key points from the Limitations section to the Introduction, providing readers with this important context upfront. We'll also include references to relevant literature from FAccT, AIES, CHI, and STS communities to place our work in the broader discourse on culture in computing context.
>
> **2. Improving annotation process description:**
> Thank you for highlighting this. We will add a concise summary of the annotation process in Section 2.2 to give a compact overview to the readers of this paper, including:
> - Image sources (participant-submitted and open-source images)
> - Key instructions given to annotators
> - A brief overview of the VQA data collection approach
> This addition will make the main text more self-contained and accessible to readers less familiar with VQA data collection methods.
>
> We appreciate your thorough review and suggestions for improving the paper's clarity and context.

---

> > ### Comment · Reviewer_nsfT · 2024-08-19
> > **Thank you!**
> >
> > Thanks for engaging with the suggestions! Good luck with the paper.

---

### Decision · Program_Chairs · 2024-09-26

**Decision:**

Accept (Oral)

**Comment:**

This is an original and potentially impactful paper in which the authors address the problem of reducing the dominance of Western culture in current datasets by creating a novel Culturally-diverse multi-lingual Visual Question Answering dataset (CVQA). In particular, the dataset does not contain only Western images. The dataset has a good size and includes data for several different languages and countries. All reviewers and myself agree that this is a great contribution to the AI community and to the conference.